

# Quantification of CO2 and CH4 emissions over Sacramento, California based on divergence theorem using aircraft measurements

Ju-Mee Ryoo[1,2], Laura T. Iraci[1], Tomoaki Tanaka[1,8], Josette E. Marrero[1,9], Emma L. Yates[1,3], Inez Fung[4,5], Anna M. Michalak[6], Jovan Tadić[6,7], Warren Gore[1], T. Paul Bui[1], Jonathan M. Dean-Day[1,3], Cecilia S. Chang[1,3]

[1] Atmospheric Science Branch, NASA Ames Research Center, Moffett Field, CA, 94035
[2] Science and Technology Corporation (STC), Moffett Field, CA, 94035
[3] Bay Area Environmental Research Institute, Moffett Field, CA, 94035
[4] Department of Earth and Planetary Sciences, University of California, Berkeley, Berkeley, CA 94720
[5] Department of Environmental Sciences, Policy and Management, University of California, Berkeley, Berkeley, CA 94720
[6] Department of Global Ecology, Carnegie Institution for Science, Stanford, CA 94305
[7] Now at Climate and Ecosystem Sciences Division, Lawrence Berkeley National Laboratory, Berkeley, CA 94720
[8] Now at Japan Weather Association, Tokyo, Japan
[9] Now at Sonoma Technology, Inc., Petaluma, CA, 94954

Submitted to

*Atmospheric Measurement Techniques*


*Correspondence to*: Ju-Mee Ryoo (ju-mee.ryoo@nasa.gov)



**Abstract**

Emission estimates of carbon dioxide ($CO_2$) and methane ($CH_4$) and the meteorological factors affecting them are investigated over Sacramento, California, using an aircraft equipped with a cavity ring–down greenhouse gas sensor as part of the Alpha Jet Atmospheric eXperiment (AJAX) project. To better constrain the emissions fluxes, we designed flights in a cylindrical pattern and computed the emission fluxes from three flights using a kriging method and Gauss's divergence theorem.

The $CO_2$ and $CH_4$ mixing ratios at the downwind side of Sacramento show relatively consistent patterns across the three flights, but the fluxes vary - as a function of different wind patterns on a given flight day. The wind variability, seasonality, and assumptions about background concentrations affect the emissions estimates, by a factor of 1.5 to 8. The uncertainty is also impacted by meteorological conditions and distance from the emissions sources. The largest $CH_4$ mixing ratio was found over a local landfill.

The importance of vertical mass transfer for flux estimates is examined, but the difference in the total emission estimate with and without vertical mass transfer is found to be small, especially at the local scale. The total flux estimates accounting for the entire circumference are larger than those based solely on the downwind region. This indicates that a closed-shape flight profile can better contain total emissions relative to one-sided curtain flight because most cities have more than one point source and wind direction can change with time and altitude. To reduce the uncertainty of the emissions estimate, it is important that the sampling and modeling strategy account not only for known source locations but also possible unidentified sources around the city. Our results highlight that aircraft-based measurements using a closed shape flight pattern are an efficient and useful strategy for identifying emission sources and estimating local and city-scale greenhouse gas emission fluxes.



## 1. Introduction

The ability to obtain accurate emissions estimates of greenhouse gases has been highlighted as an important issue for many decades, not only for regulating local air quality but also for assessing national-scale air quality and greenhouse gas emissions. In particular, urban emissions need to be well-understood because approximately 70 % of anthropogenic greenhouse gas emissions originate from urban areas (International Energy Agency, 2008; Gurney et al., 2009, 2015). This often gives rise to *urban domes* with higher greenhouse gas (GHG) mixing ratios than

surrounding areas (Oke, 1982; Idso et al., 1998, 2002; Koerner and Klopatek, 2002; Grimmond et al., 2004; Pataki et al., 2007; Andrews, 2008; Kennedy et al., 2009; Strong et al., 2011). Therefore, estimating greenhouse gas emissions at regional to national scales requires an improved understanding of urban GHG emissions and the role of human behavior in altering these emissions (Rosenzweig et al., 2010; Wofsy et al., 2010a, b).

        The commonly used *bottom–up* inventories derive estimates of direct and indirect emissions of greenhouse

gases based on an understanding of emission factors from the constituent sectors (Andres et al., 1999; Marland et al., 1985; Boden et al., 2010; California Air Resources Board, 2015; US EPA, 2016). These estimates rely on monthly or quarterly statistical averages of emission activities and often time-invariant emission factors, which mask behavioral patterns. However, recent *bottom–up* inventory data have improved from coarse estimates by using proxy data to produce fine spatial resolution estimates using specific activity data and emission factors corresponding to

each emission source. In contrast, *top–down* methods (or inverse modeling), in which observed mixing ratios are partitioned into their sources, have also been used for constraining or cross-checking bottom-up emissions (Huo et al., 2009; Zhang et al., 2009; Cohen and Wang, 2014; Fischer et al., 2016; Miller and Michalak, 2017).

        Efforts to understand urban-scale emission using direct observation have been undertaken in several large cities including the Northeastern U.S. (Boston, Baltimore/Washington D.C., He et al., 2013; Dickerson et al., 2016),

the U.S. Mountain West (Salt Lake City, Strong et al., 2011), Indianapolis (Mays et al. 2009; Turnbull et al., 2015; Lamb et al., 2016; Lauvaux et al., 2016), and the Southwestern U.S., especially the Los Angeles basin (Duren et al., 2011; Kort et al., 2012). There are several methods to quantify emissions: in-situ measurements and flask collection through surface tower systems, space-based satellite retrieval, airborne in-situ measurements, mesoscale models, and Large Eddy Simulation (LES) modeling. As part of the Indianapolis Flux Experiment (INFLUX) project, airborne

and tower measurements have been collected throughout Indianapolis to generate an extensive database. Over the western U.S., a legacy dataset over Salt Lake City has collected measurements of $CO_2$ using surface tower systems for more than one decade (Pataki et al., 2005, 2007; Strong et al., 2011). Results from this extensive dataset have included seasonal variability over years and source apportionment into anthropogenic and biogenic sources. While those efforts reach general agreement on emission inventories across the cities, they have limitations due to their

geographical differences in topography, climatology, different source attributions (such as types of industry and agriculture), as well as differences in the measurement and analysis methods.

        One approach for estimating $CO_2$ and $CH_4$ fluxes over cities is the use of an aircraft-based mass balance method. Several studies have demonstrated the utility of this approach (Kalthoff et al., 2002; Mays et al., 2009; Turnbull et al., 2011; Karion et al., 2013, 2015; Cambaliza et al. 2014; Gordon et al., 2015; Tadić et al., 2017). Mass



balance methods utilize many length scales and patterns. The flights target mostly local scales (< 3 km) and areas around the point sources (Nathan et al., 2015; Conley et al., 2017), but they also characterize urban-scales (e.g. 25 x 10 km for Gordon et al. (2015), 4 x 9 km for Tadić et al. (2017)) and the large-scale (40 km up to 175 km, especially for a downwind curtain flight (Mays et al., 2009; Turnbull et al., 2011; Karion et al., 2015)).

The flight patterns can be classified into three different categories: 1) single-height transect flight, 2) single
screen ("curtain") flight with multiple transects, and 3) enclosed shapes (box, cylinder) (see Fig. S1 in Supplementary Appendix). Commonly, there are assumptions made in these airborne sampling approaches. First, the single-height transect approach assumes a well-mixed boundary layer. Karion et al. (2013) measured $CO_2$ and $CH_4$ along a single-height transect with an assumption of uniform vertical mixing. Turnbull et al. (2011) performed a flux estimate by incorporating detailed meteorological information and transecting an emission plume with an aircraft.
These studies also assumed that emissions originate from point sources such as pipes and smokestacks, and travel downwind so that all pollution is reflected on the downwind "curtain" with constant wind speed. Second, the single-screen multi-transect method does not assume a uniformly mixed boundary layer condition but is dependent upon a constant wind speed. Without a well-mixed boundary layer assumption, Cambaliza et al. (2014) measured $CH_4$ along multiple height transects downwind of the city of Indianapolis (See Fig. S1a in Supplementary Material).
However, they assumed that winds at the time of measurement were the same as at the time of emission (i.e., winds after the methane release were time-invariant). Third, the enclosed 3-D shape flights do not presuppose any of the assumptions described above. Gordon et al. (2015) measured various GHG with a stacked box flight pattern, to capture the vertical variation in mixing ratio both upwind and downwind. Tadić et al. (2017) and Conley et al. (2017) accomplished emission estimates by flying a cylinder pattern near an emission source to measure GHG both
upwind and downwind for analysis based on the divergence theorem. More recently, Baray et al. (2017) used both a screen flight and box flight approach around oil sands facilities and showed that each flight pattern could be preferred, depending on the types of emissions and spatial characteristics.

While these assumptions may be valid in certain conditions, they do not always hold. Most cities include multiple sectors, including industry, agriculture, and residential areas, and can have daily variability in wind that
influences flux patterns. For example, a local landfill, various highways and several airports around Sacramento are significant sources of emissions and of uncertainty. To minimize uncertainty in flux estimates, these sources must be taken into consideration. The method of extrapolation to unsampled areas can also be a large source of uncertainty. For example, Gordon et al. (2015) demonstrated the significant impact of extrapolation methods over the unsampled, near-surface region on the final emission estimate, unlike Cambaliza et al. (2014) who assumed that the city plume
is rarely observed in a transect between the surface and the lowest altitude flight measurement. Assumptions can break down when wind direction and speed vary with time and three dimensional space (e.g. longitude, latitude and altitude; see Fig. 2S); incorrect use of wind data can result in increased uncertainty and reduction of accuracy. Flux estimates also require an estimate of the planetary boundary layer height (PBLH), an important physical parameter that is hard to measure. State-of-the-art atmospheric models and reanalysis products often estimate the PBLH, but
substantial differences have been observed in existing models and reanalysis data (Wang et al. 2014). In addition, entrainment from the free troposphere into the planetary boundary layer (PBL) and fluxes from the surface have



been ignored in most previous studies. Thus, more careful consideration and understanding of these factors are required for determining emission estimates using any of the three mass balance flight patterns.

The primary goals of this study are: i) to design and execute the cylindrical flight patterns for greenhouse gas
observations for an urban domains (Fig. S1b in Supplementary Material) and to assess the impact of different interpolation and extrapolation methods on the emission estimate, ii) to test the sensitivity of emission estimates to a variety of factors such as wind, background mixing ratios, and different flux estimation methods, and finally iii) to examine the importance of vertical mass transfer on the flux estimates. To address these goals, we collect $CO_2$ and $CH_4$ data during three flights over Sacramento (See Fig. S3 in Supplementary Material) for urban (35–60 km) and
local scales (< 3 km), and determine emission fluxes using various treatments of wind conditions, background value, and vertical mass transfer. The data and methodology are presented in section 2. Kriging results for the $CO_2$ and $CH_4$ concentrations fluxes for all three flights are shown in section 3. The sensitivities of flux estimates to different use of the wind, background value, and vertical mass transfer are also investigated. The conclusions of this study are presented in section 4.

**2. Data and Methods**

**2.1 Data Collection**

In situ measurements of $CO_2$ and $CH_4$ were performed as part of the Alpha Jet Atmospheric eXperiment (AJAX) project. The $CO_2$ and $CH_4$ instrument (Picarro Inc., model 2301-m) is calibrated before each flight using whole-air standards from the National Oceanic and Atmospheric Administration's Earth System Research
Laboratory (NOAA/ESRL). Water vapor corrections using Chen et al. (2010) were applied to calculate the dry mixing ratios of $CO_2$ used during this study. The overall uncertainty was determined to be 0.16 ppm for $CO_2$ and 2.2 ppb for $CH_4$ (Tanaka et al.,2016; Tadić et al., 2014).

The Meteorological Measurement System (MMS) measures high-resolution pressure, temperature, and 3-D ($u$, $v$, and $w$) winds (Hamill et al., 2016). Take off time from Moffett Field was between 20:30 and 21:00 UTC for each
flight day discussed here. For the November flights, local time was 13:30-14:00 Pacific Standard Time; for the July flight, take off was at 12:37 Pacific Daylight Time.

**2.2 Extrapolation to the Surface**

Because the lowest flight path was typically 250–380 m above the surface and there were no ground-based measurements along the flight tracks, there is always a gap in measurement data between the surface and the lowest
flight altitude. Many studies adopt a well-mixed layer assumption below the lowest flight altitude (Karion et al., 2013), but these unmeasured values can lead to a significant bias and large uncertainties in estimating GHG mixing ratios and fluxes, depending on interpolation and extrapolation schemes (Gordon et al., 2015). Thus, we investigated four methods to extrapolate mixing ratio values to the surface, which are termed 1) constant, 2) exponential fit, 3) Gaussian fit, and 4) kriged fit. The constant method assumes an elevated plume with a constant background level.
The background level is derived from the lowest flight measurement: $X(t, z) = X(t, z_L)$ for $z_0 < z < z_L$, where $z_L$ is the lowest flight level. The exponential-fit method assumes an exponential decay of $X(t, z)$ from $z_L$ to $z_0$. The



detailed method is based on Gordon et al. (2015). The Gaussian fit method is similar to exponential fit method, except that the surface-sourced plume dispersion follows a Gaussian distribution. The kriged fit was applied down to the surface level, extended from the sampled area above.

**2.3 Elliptical Fit and Kriging**

Because the aircraft flew in a cylindrical pattern around the city, the flight paths were transformed into a polar coordinate system. The path was projected to the surface first and fit into an ellipse using least squares method to minimize the difference between the measured data and the fitted data. Then, we computed each point using the major and minor axis of the ellipse and parameter $t$. Each point on the ellipse was represented by a single parameter
($t$, eccentric anomaly), according to the equations:

$$X(t) = X_0 + a\cos t\cos\phi - b\sin t\sin\phi$$
$$Y(t) = Y_0 + a\cos t\sin\phi + b\sin t\cos\phi$$
(1)

where $a$ and $b$ is a radius of the major, minor axis of the ellipse, respectively, $\varphi$ is the angle between the X-axis and
the major axis of the ellipse, and the parameter $t$ is obtained from Eqn. (1), varying from 0 to $2\pi$. Then, the data was gridded into a two-dimensional plane [$t$, height].

In order to assess the strengths of a kriging approach to quantifying emissions, two interpolation methods were assessed: interpolation using kriging and that using an exponential weighting function. The exponential weighting function at a given point ($P$) was defined as the weighted average of all the other points where the weights decrease
exponentially with distance to $P$. Both approaches captured the general plume pattern (regions with high and low concentrations of $CO_2$), but the kriging approach did better at capturing individual plume features such as the range and magnitude, while interpolation with the exponential weighting function could not resolve such details (see Fig. S4 in Supplementary Material). Another benefit of kriging is that it can estimate values at unsampled locations using a weighted average of neighboring samples, thus reproducing the characteristics of the observed values.

Interpolation was performed by the ordinary kriging method (Chilés and Delfiner, 2012), modified from the IDL v8.1 kriging tool to fit an elliptical pattern. We chose ordinary kriging because there is no obvious trend in the data we use. Before kriging, we modeled the variograms for all relevant variables. A variogram (or semivariogram) is a function describing the degree to which the data are correlated as a function of the separation distance between observations. The empirical semivariogram of the data was fit using an exponential variogram model, based upon
visual inspection of the experimental variograms. Three parameters were used to fit the theoretical variogram, namely the sill (the expected value of the semivariance between two observations as the lag distance goes to infinity), the range (the distance at which the variogram reaches approximately 95% of the sill), and nugget (representative of measurement error and amount of microscale variability in the data). Variogram modeling was first performed to derive parameters required to obtain ordinary kriged estimates. Various other types of kriging
exist in the literature on quantifying greenhouse fluxes (Tadić et al., 2017), but examining their differences is beyond the scope of this study.





We kriged the $CO_2$, $CH_4$, wind, temperature, and pressure observations to obtain both the estimate and the uncertainty for each variable at each grid point. The individual semivariograms of the variables for each flight were produced, and we present them for one flight in Fig. S5 in Supplementary Material. For each flight, the sampled data were kriged to a grid of maximum height divided by 150 in the vertical dimension; the horizontal dimension was kriged from end to end of the flight transect, enclosing the circumference of the entire city, divided by 360 in the horizontal direction. The vertical dimension was interpolated from the ground to the top of the flight measurement, but only data up to the estimated planetary boundary layer height (PBLH) was used for computing the flux estimates.

The uncertainty in the kriged results was assessed using the variance (and the standard deviation) of the kriged estimate at each point, as in Mays et al. (2009) and Nathan et al. (2015). In a statistical sense, the interpolated downwind $CO_2$ and $CH_4$ concentration is one of the largest sources of uncertainty in flux error estimates because the downwind flux calculation requires interpolated values at unsampled locations. Another well-known significant source of uncertainty comes from the wind measurement (Mays et al., 2009; Karion et al., 2015; Tadić et al., 2017). The grid resolution can also be a source of uncertainty. Nathan et al. (2015) reported that changing the grid resolution by a factor of 2 in either direction resulted in a 4 % absolute change in the emission rate, and showed that the grid size does not significantly bias the interpolated emission rates for their study. However, emission estimates may depend on scales of variability in the measured quantities and the grid resolution, in that the grid resolution has to be sufficiently fine to capture the observed scales of variability. They also demonstrated that the selection of the variogram model they used, such as Gaussian-cosine, linear, exponential, and exponential-bessel variogram, did not affect the final emission estimate substantially (the difference is less than 5 %) in their case study. Moreover, uncertainties in greenhouse gas mixing ratio measurements, as well as in wind speed and direction, directly propagate the emission rates uncertainties.

### 2.4 Treatment of Measured Wind

To test the sensitivity of fluxes to the choice of the background value and the wind characteristics, we applied the measured high-resolution (1 Hz) in-situ wind data to the flux calculation in two different ways. In one we used the measured wind at specific measured points without any correction (hereafter we refer to it as "raw wind"). In this case, inflow and outflow are not balanced within a cylinder. Alternatively, we averaged horizontal wind at each vertical level, so that air (mass) coming into the cylinder equaled air leaving the cylinder. By assuming non-divergence, mass can be balanced (here we refer to it as "mass–balanced mean wind").

### 2.5 Planetary Boundary Layer Height (PBLH) and Entrainment

The potential temperature profile, which indicates atmospheric static stability and which significantly affects pollutant diffusion, is the most common operational method to determine PBLH. We determined PBLH as the altitude of the maximum gradient from a vertical profile of potential temperature (Wang et al., 2008) obtained from the MMS measurements. No significant sensitivity was found using several different PBLH detection algorithms, such as the parcel method (the interaction between dry adiabatic lapse rate and temperature), rapid decrease in water



vapor (Wang and Wang, 2014), or Richardson number method (Wang et al. 2008). A simple example is shown in Supplementary Material Fig. S6.

The boundary layer growth is determined by the sum of entrainment velocity ($w_e$) and large-scale mean vertical velocity (Vilà-Gueru de Arellano et al. 2004, Faloona et al., 2005, Trousdell et al., 2016). Here we define the entrainment (surface) flux as the turbulent flux of the scalar at the boundary layer height (surface). Then we compute the entrainment flux at the top of the cylinder by multiplying the area of the top of the cylinder with $E = \overline{(w'c')}_h \cdot A$ where $A$ is the area of the top of the cylinder, $c' = (C(t,z) - C_{bg}(h))$, $C(t,z)$ is the $CO_2$ mass (g m$^{-3}$) at a given point surrounding the top of the cylinder at $z = h$, and $C_{bg}(h)$ is the background concentration of $CO_2$ at the top of the boundary layer. The $CO_2$ mass is calculated from the $CO_2$ mixing ratio (ppmv). Using this, we could make direct observations of the entrainment flux by measuring vertical velocity together with the trace gas mixing ratio throughout the boundary layer. The surface flux is computed at the surface (z = 0) in a similar manner.

## 2.6 Background Values, Flux Calculation and Emission Rate

Figure 1 shows a map of the AJAX flight tracks over Sacramento and the vertical structure of the $CO_2$ mixing ratio on November 17, 2015. A simple illustration of the air flow (computed as the pressure divided by the magnitude of the wind vector normal to the cylindrical surface) demonstrates the basic idea of this study, Gauss's divergence theorem, which relates the flow through the surface to the flow inside the surface. Mass coming in and out of the cylinder should be conserved if there is no leak throughout the top and the bottom of the cylinder (i.e., the flow into the cylinder balances with the flow out of the cylinder). More precisely, the outward flux of a vector through a closed system is *equal* to the volume integral of the divergence over the region inside the surface. Since the atmosphere has no upper boundary, we assume that vertical mass transfer is accomplished through an entrainment from the top of the PBL, and surface flux from the bottom of the cylinder near the surface. In this way, the oval cylinder we design over the city has a closed surface, and the flux inside the cylinder is equal to the sum of the emission flux at the bottom.

Background values are one of the most important factors in obtaining flux estimates. Here we used three distinct methods to determine background values and calculate emission fluxes for each gas. First, we used the *minimum* value at each vertical level of the data (i.e., from the surface to the top of the PBL). Second, we used the *average* value at each vertical level. Third, we used two different vertically invariant, constant values throughout the whole height from the surface to the top of the PBL.

We determined kriged data for each field from the measured $CO_2$, $CH_4$, wind, temperature, and pressure, and then estimated the local background concentrations for both trace gases following three different approaches described above. Then we subtracted these background values from the trace gas data at each grid point. To convert the volume mixing ratio [ppmv] to a mass concentration [g m$^{-3}$], the number of $CO_2$ or $CH_4$ molecules were computed based on the ideal gas law using the kriged temperature and pressure. Then, the net mass flow [g m$^{-2}$ s$^{-1}$] was integrated in the horizontal and vertical directions from the surface up to the top of the cylinder.



$$F = \iint \overline{U}(\theta, z) \sin(\alpha) \cdot (C(\theta, z) - C_{bg}) L \, d\theta \, dz \qquad (2)$$

where $L$ is the difference between two points on the ellipse, $\overline{U}(\theta, z)$ is the wind speed, $\alpha$ is the angle of the wind

velocity relative to the flux surface, $C$ is the concentration (g m$^{-3}$), and $C_{bg}$ is the background concentration at each

level z. The component of the wind perpendicular to the flux surface was used in the flux calculation.

**3. Results**

**3.1 Dependence of GHG mixing ratios on different interpolation schemes and extrapolation fits**

Estimates of GHG mixing ratios along the cylindrical boundary of the flight pattern are sensitive to the choice
of interpolation schemes and extrapolation fits, especially at lower altitudes where there are no aircraft data available.

Figure 2 shows $CO_2$ and $CH_4$ mixing ratios for November 18, 2013 and November 17, 2015 at several locations over
Sacramento. These results demonstrate that a large source of uncertainty and difference comes from not only the
interpolation but also the extrapolation of the data between the lowest flight level and the surface. For example,
uncertainty in estimated GHG mixing ratios below the lowest flight level (indicated by the yellow diamond) can be
large (up to ~ 20 %). In the worst cases, $CO_2$ mixing ratios span more than 60 ppm at the surface among the

methods; $CH_4$ ranges > 0.15 ppm. Note that the differences between interpolation schemes where data exists (above
~ 250–380 m) were smaller than the differences between the values obtained from the various methods below the
lowest flight data. Without ground-based data, a proper choice of extrapolation schemes requires knowledge or
presumption of the mixing ratio behavior in this region. Gordon et al. (2015) proposed that the case of elevated
sources beneath the lowest flight level is best suited to constant extrapolation of mixing ratio to the surface (blue

curve), while a ground-source concentration should be represented with an exponential-fit extrapolation (red).

The various fits rely on different assumptions; the ordinary kriging method (magenta traces in Figure 2) also
requires some assumptions (e.g., constant mean, constant variance, second-order stationarity and isotropy, and
validity of the theoretical model). Ordinary kriging leverages spatial and statistical properties of the observations to
derive estimates, and seems to be less arbitrary than alternative interpolation/extrapolation methods. We note the

similarity between the kriged values and the constant extrapolation method for both $CO_2$ and $CH_4$.

**3.2 Observed and Kriged GHG Mixing Ratios**

Figure 3 shows the measured methane over Sacramento, CA, for November 18, 2013, and the projection to the
ground (plan-view), the grid in panel (d) shows how we fit the data to compute the "flux surface" using kriging. The
kriged $CH_4$ not only captures the measured $CH_4$ mixing ratio, but also fills the gap of the unsampled area based on

the observed data characteristics. Maximum values were found at 38.73° N, 121.2° W to 38.68° N, 121.45° W at 300
m. The high $CH_4$ region corresponds with highways, airports and dairy farms, especially near the surface. Compared
to the conventional interpolation without considering the individual characteristics of data, kriging can capture the
most important features of the data (see Fig. S4 in Supplementary Material).



Figure 4 shows the observed and kriged $CO_2$ mixing ratio, and kriging uncertainty at each grid point on November 18, 2013, and November 17, 2015. The kriged $CO_2$ fields capture the main features of the observed $CO_2$ plume well. The centers of the large $CO_2$ plumes differed somewhat in magnitude and width, reflecting the varied source characteristics for $CO_2$. The $CO_2$ mixing ratios on November 18, 2013, were much larger (up to 25 ppm higher at most spots) than those on November 17, 2015.

The observations in Figs. 4(a–f) suggest that the vast majority of the emission sampled by the flights originates in the region identified as traffic regions (Roseville), airports, metropolitan areas (Arden-Arcade-Roseville, Fair Oaks, North Highland), and dairy farms. On both days, the largest $CO_2$ mixing ratios were seen on the northeast side of the oval, which was the downwind side when sampled in 2013 and the upwind side in 2015 (see Fig. S2 in Supplementary Material), indicating that all areas need to be taken into consideration to understand the emission characteristics, obtain the actual flux estimates, and reduce uncertainties. The detailed wind direction speed are shown in Fig. S2 in Supplementary Material.

The vertical stretching pattern of $CO_2$ mixing ratios in Figs. 4(g-j) appears to be due to the large scale difference between the horizontal length (> 120 km) and the vertical length (< 1 km). When we applied our method to the local scale (horizontal scale < 3 km, see Fig. 6), or took a small horizontal portion of the large loop (see Fig. S4 in Supplementary Material), the vertical stretch pattern disappeared.

Panels (i, j) show the uncertainty was largest near the ground, in particular over the unmeasured area (e.g., November 17, 2015, below ~ 200 m, 38.63º N, 121.11º W ~ 38.81º N, 121.18º W), and when the data were observed a farther from the elliptical path. The uncertainty in a narrow region at ~1 km for November 17, 2015, resulted from the lack of data when aircraft was entering and exiting the cylinder. Furthermore, there were no measurements on the ground, so the estimates below 200 m were dependent only on the data around 200 m, which was an additional source of the uncertainty. Uncertainties of $CO_2$ were large near the surface, small from 200–900 m, and grew larger near the top of the sampled domain.

By assuming that the errors of each factor are Gaussian in nature and each measurement (e.g. $CO_2$ and wind) is independent (no covariance), we estimate the overall uncertainties in the calculated flux by adding the fractional uncertainties of the kriged $CO_2$, $CH_4$, and winds in quadrature, as in Nathan et al. (2015). The overall uncertainties of the $CO_2$ and $CH_4$ mixing ratios (emission estimates) are similar: both of them are less than 2(1) %. When we only compute the fluxes for the lateral part of the cylinder using the errors from the $CO_2$ and $CH_4$ mixing ratios and the wind measurement (normal component of the flux surface), the overall uncertainty of the emission estimate over the urban scales is about 4%, and 14% for both $CO_2$ and $CH_4$ on November 18, 2013 and November 17, 2015. The uncertainties over the local scales over landfill and rice field for both $CO_2$ and $CH_4$ on July 29, 2015 are about 35% and 17%. When we include the entrainment flux at the top and the surface flux at the bottom of the cylinder in addition to the flux in the lateral part, the total uncertainty was increased by about less than 1% or remains the same for both $CO_2$ and $CH_4$. This appears to be because the contribution of the vertical mass transfer through entrainment and the surface flux to the total flux estimates is relatively small, as we will show in section 3.5.

Although we used much more accurate in-situ wind measurements than most past studies for flux calculation, the wind was still the most important variable for the uncertainty of flux estimates, consistent with previous studies.





This probably partially stems from the uncertainty in the wind at interpolated locations or the sparsity of the measurements. Cambaliza et al. (2014) estimated the uncertainty of the emission rates from kriging analysis is about 50 %. Nathan et al. (2015) also estimated the overall statistical uncertainty of the emission rate over a compressor station in the Barnett Shale as ± 55 %. We did not consider the uncertainty of PBLH here because we used the

PBLH based on our in-situ meteorological data and estimate its uncertainty to be < 1 %.

### 3.3 Sensitivity of Calculated Flux to Wind Treatment

Wind variability and measurement assumptions can lead to errors in the $CO_2$ and $CH_4$ flux estimates (Mays et al., 2009; Cambaliza et al., 2014, 2015; Nathan et al., 2015; Karion et al., 2013, 2015), and the way in which winds are estimated and quantified especially matters. For November 18, 2013, the wind was southwesterly at the low

altitude, but it changed its direction to southeasterly as height increased. Figure 5 demonstrates the clear difference in flux estimates when the 2-D raw wind and the mean for each level (mass balanced) wind are used. Note that we captured high fluxes (panel f) along with high $CO_2$ and $CH_4$ mixing ratio when we used the mass-balanced wind (e), while we were less likely to obtain a strong emission signal when using the raw wind data (h), which might be attributed to an imbalance of inflow and outflow to the cylinder. The total flux was ~7 times different between wind

cases: 3.68 Mt $CO_2$ yr$^{-1}$ and 13.00 Gg $CH_4$ yr$^{-1}$ calculated with raw wind, and 26.55 Mt $CO_2$ yr$^{-1}$ and 88.82 Gg $CH_4$ yr$^{-1}$ with mean wind using the minimum mixing ratio of each level as the background and including vertical mass transfer (see Table 1, rows 1 and 3).The difference was much less for the flight in November 2015.

The importance of wind data on the flux calculation is also seen in local–scale emission calculations (see Fig. S8 in Supplementary Material). For the small cylinder over the landfill site on July 29, 2015, Fig. 6 shows the

observed and kriged $CH_4$ mixing ratio and the flux estimation using either the raw wind (c) or the mass-balanced wind (d). As before, the kriged $CH_4$ is a good representation of the local characteristics of the $CH_4$ field. Reassuringly, the elevated $CH_4$ concentration was reconstructed over 121.19º W, 38.52º N, which was the actual location of the landfill (See also Fig. S8 in Supplementary Material). The $CH_4$ fluxes (7 – 10 Gg $CH_4$ yr$^{-1}$) estimated using measured wind (varying with time and space) reflect the strong enhancement of $CH_4$ mixing ratios at local

scales. Considering light wind conditions (< 2.5 m s$^{-1}$) and high temperature during July, the local emissions are attributed to these high flux estimates. The choice of distinct wind treatments led to a 30% difference in the total $CH_4$ estimate and 70% in $CO_2$ for this particular case. The $CH_4$ concentration and its flux is low over the rice field on that day (See in Fig. S7-S8 in Supplementary Material).

Many previous studies estimated $CO_2$ and $CH_4$ fluxes based on the mean wind vector at the dominant wind

direction (positive and one direction) and speed (Turnbull et al. 2011; Karion et al. 2015), often using simulated wind obtained from a coarse resolution model. The mass-balanced area-mean wind of the cylindrical loops in this study was based on actual measurements, not coarse resolution model data, which enhances the accuracy of our flux estimates.


**3.4 Sensitivity to the Choice of Background Concentrations**

Since both $CO_2$ and $CH_4$ mixing ratios vary with altitude, we employed a vertically variant background value for each trace gas. Tables 1 and 2 show the calculated $CO_2$ and $CH_4$ emission fluxes using two different wind methods and two different background treatments for different flight days. The rows labeled "min" were generated using the minimum kriged mixing ratio in each altitude band as the background for all data at that level. The rows in Table 1 identified by "Bg=avg" used the average mixing ratio on each of the 150 vertical levels as the background

on that level.

The sensitivity of calculated flux to the choice of the background treatment was significant when we used raw wind (top two rows in Tables 1). This was true both with vertical mass transfer (Table 1) and without (not shown). In contrast, when we use the mean wind, the emission estimates for both $CO_2$ and $CH_4$ are nearly identical for either choice of the background treatment.

Interestingly, emission rate estimates were similar for the case of raw wind with averaged $CO_2$ or $CH_4$ as the background values and both cases using mass-balanced (mean) wind. To satisfy mass conservation, we also computed the entrainment flux from the top ($z=h$) and the surface flux from the bottom of the cylinder ($z=0$). The data from Table 1 is also shown in Figure 7 as the non-hatched bars.

Our city-wide estimate of 15-28 Mt $CO_2$ $yr^{-1}$ is higher than the result by Turnbull et al. (2011), who reported

3.5 Mt $CO_2$ $yr^{-1}$ over Sacramento in February 2009. When we examine only the small portion of the ellipse which shows the highest $CO_2$ mixing ratio (e.g. 121.45–121.20° W and 38.65–38.76° N in 2013), $CO_2$ fluxes calculated using spatially varying wind with minimum values for background were 4.2 Mt $yr^{-1}$ in 2013 and 5.5 Mt $yr^{-1}$ in 2015. When calculating fluxes using mean wind with average values for background, the "downwind side" emission rates were about 4.4 and 3.5 Mt $yr^{-1}$. From this study, the fluxes from the downwind portion of the cylinder were

responsible for only ~15–23 % of the total emissions.

Our city-wide estimate of 89-95 Gg $CH_4$ $yr^{-1}$ on November 18, 2013 corresponds to 53-57% of the 167 Gg $CH_4$ $yr^{-1}$ (about 140–220 Gg $yr^{-1}$). reported by Jeong et al. (2016) over Region-3 (San Joaquin Valley area including Sacramento). On November 17, 2015, we calculate a significantly smaller emission rate (8 – 11 Gg $yr^{-1}$). Direct comparison between different flux estimates is challenging due to various factors, such as i) differences in the areas

covered, ii) differences between bottom-up inventory and top-down estimates, iii) the variance of measurement methods (tower, aircraft, and model), iv) underestimation of the emissions from known sources, v) seasonal and interannual variability, vi) different spatial coverages, and vii) lack of understanding of unidentified sources. This will be one of the most important areas for improvement for establishing better emission estimate databases in the future.

**3.5 Importance of Including Vertical Mass Transfer**

Many previous studies assume that vertical mass transfer can be neglected (Cambaliza et al., 2014; Conley et al., 2017). To quantify the validity of this assumption, we compare the flux determined when including or neglecting the entrainment and surface fluxes. Figure 7 shows the urban-scale emission rate estimates over Sacramento, CA using spatially and temporally varying ("raw") wind and the mean ("mass balanced") wind. We chose the





background values as i) average value of each vertical level, ii) the minimum value of each level, or iii) one of two fixed values at all altitudes. The fixed values were chosen in the range between the minimum and median mixing ratios. Note that the total fluxes using the mean wind were not sensitive to the choice of the background value (< 3 %, whether background value was minimum, average, or constant values). However, as discussed in Section 3.4, the total fluxes using spatially varying wind (u, v) were sensitive to the choice of the background value on both
flight days for both gases. Like the urban scale analysis, the flux estimates over the landfill and rice field local scale cylinders using raw wind with average background were similar to those using mass-balanced mean wind at each level with either minimum or average values as background (shown in Table 2). The differences in $CO_2$ and $CH_4$ fluxes with and without (hatched bars in Figure 7) vertical mass transfer were determined to be only 16-17% on the urban scale and less important for the local emission estimates (< 10 %).

**4. Conclusions**

We have estimated $CO_2$ and $CH_4$ fluxes over Sacramento, California, on three days using an airborne in-situ dataset from the Alpha Jet Atmospheric eXperiment (AJAX) project and have tested the sensitivity of emission estimates to a variety of factors. First, we deployed cylindrical flight patterns of two sizes that differ from common curtain flights to estimate the total flux at urban and local scales. We also applied a kriging interpolation method to
the data, capturing the characteristics of the data at both observed and unsampled locations. Second, we tested the sensitivity of flux estimates to the wind, background concentrations, and different flux calculation methods. We found that the way in which winds are estimated and how background values were chosen were the dominant factors in determining the total flux estimate. Third, we took into account not only the inflow and the outflow through the cylinder around the city, but also the vertical mass transport (e.g., entrainment and surface flux) and tested the
sensitivity of the total flux estimate to the vertical mass transfer for both urban and local scales.

When we used the area-mean wind for flux calculation, the sensitivity of the emission estimate to the choice of background was minimal (Table 1). Urban scale $CO_2$ flux was similar in both years sampled (20-27 Mt $CO_2$ yr$^{-1}$), but the calculated $CH_4$ flux was different by ~9x (~90 Gg $CH_4$ yr$^{-1}$ in 2013 and ~10 Gg $CH_4$ yr$^{-1}$ for the flight in 2015). Using measured winds, not averaged, produced similar flux estimates when the background mixing ratio was
set to the average value on each vertical layer. In contrast, choosing the background as the minimum value observed on each level led to calculated fluxes that were substantially lower for the flight in 2013.

The Planetary Boundary Layer Height (PBLH) was calculated using the vertical profiles of potential temperature and was used together with the vertical fluxes for computing the entrainment from the top and the surface flux from the bottom of the cylinder. Neglecting vertical mass transfer can increase the uncertainty of the
total flux estimate by up to 17 % in our cases.

The advantage of the closed shape (i.e., elliptical in this study) approach over the curtain flight is to make a more precise "total" emissions estimate possible by taking into account all unknown sources of emissions. Regarding the balanced incoming and outgoing fluxes within a closed volume, we suggest that emission estimates using mean measured wind computed over a closed shape can be beneficial for several reasons. First, the flux
estimates calculated using mass-balanced mean wind reduce the sensitivity to the choice of background. From Fig. 6



and Table 1 we found that the background value is one of the major sources of uncertainty in both $CO_2$ and $CH_4$ emission estimates, but when we use as the background value either the minimum value (which is often similar to the air away from the source) or the mean value at each vertical level, the final flux estimates become similar. Second, when we analyze only a small portion of the large loop (e.g., downtown hot spot region) to mimic the curtain flight style, the final flux estimates are highly sensitive to the background choice no matter how the measured wind data are treated. These also indicate that the flux estimates for the closed elliptical loops over the city would reduce sensitivity to the choice of background values.

The local $CH_4$ flux over a landfill is one-tenth to one-third the total emission flux over Sacramento in wintertime, suggesting a local point source of $CH_4$ can play a significant role in enhancing overall $CH_4$ concentrations. In general, $CH_4$ over a rice field showed lower emission rates than those over the landfill, and this may be due to the relatively high wind, no particular point source, and reduced $CH_4$ emissions as a result of low humidity (dry) conditions. Considering the wind speed was much lower in July (especially over the landfill), this indicates that most of the emission was produced from local sources for the July 29, 2015, case. On the other hand, the strong winds observed on November 18, 2013 came from the southeast, isolating high concentrations of $CO_2$ downwind of industrial facilities. On November 17, 2015, stronger upper level wind came from the north, but the weaker lower level wind originated from the south. High concentration of $CO_2$ at lower altitudes were thus located downwind of a major highway intersection and residential sectors. Elevated $CH_4$ concentrations were found over local, narrow, and isolated areas for both November flights.

The spatial variation of $CO_2$ and $CH_4$ observed in the cylindrical flight pattern measured over Sacramento reveals that there were several local sources throughout the entire city, not only concentrated on the downwind side. Furthermore, variability of wind may contribute to different flux estimates at a similar time of year (e.g., November). Our sensitivity study reveals that the unbalanced wind varying with time and space may be a source of methodological uncertainty. Thus, use of constant wind speed or unrepresentative coarse resolution of wind (e.g., model output with coarse resolution) by focusing only on the downwind side may lead to significant uncertainty in the estimation of the greenhouse gas emission fluxes. The size of the ellipse measuring urban emission appeared to be another factor affecting flux estimates. In general, vertical mass transfer does not significantly contribute to the total emission estimate (especially at local scales), but it can modify total emission estimates by up to 17 % for urban scales in our cases. For the local scale (~ 3 km), vertical mass transfer was not important due to the small turbulent fluxes.

There are still several issues to be addressed further. First, further sector-specific emissions and their uncertainties for $CO_2$ and $CH_4$ need to be further identified (Miller and Michalak, 2017). Second, the seasonality of sensitivity of emission estimates to various factors needs to be examined. Finally, understanding the sources of uncertainties in emission estimates, and how different they can be under various conditions need to be investigated further. We found that the uncertainty of the emission estimates can be also sensitive to temperature, and potential temperature (to determine PBLH). In this sense, the changing climate over California makes it harder to predict future emission patterns. The use of aircraft measurements presented here provides the tremendous opportunity to measure the entire urban plume.



This effort is not limited to one particular city. There has been increasing interest in performing inter-city comparisons to validate datasets in a more efficient and adequate manner, to create a uniform database that is useful

for emission controls (Urban greenhouse gas measurements workshop, 2016). Given that data are available over several cities which have different conditions, we can test how to obtain emission estimates from several cities. Differences in the socio-economic, geologic, and industrial characteristics of cities lead to a need to compare emission estimates between them, as together they can contribute significantly to the total GHG emission at national and global scales. Thorough comparison among datasets and a customized sharing system between different

research groups will lead to reducing the uncertainty of emission estimates.



**Author contribution**

Ju-Mee Ryoo, Laura T. Iraci, Tomoaki Tanaka, Josette E. Marrero, Emma L. Yates, Warren Gore designed the experiments, and they carried them out. T. Paul Bui and Cecilia S. Chang prepared MMS instruments on Alpha jet
and Jonathan. M. Dean-Day processed the data. Anna M. Michalak and Jovan Tadić participated in experiment design and supported the interpretation of the statistical analysis. Inez Fung gave insightful comments, and Laura T. Iraci gave helpful guidance in formulating the structure of this study. Ju-Mee Ryoo developed the statistical model code and performed the analysis as well as prepared the manuscript with contributions from all co-authors.



**Competing interests**

The authors declare that they have no conflict of interest.



**Acknowledgements**

The authors appreciate the support and partnership of H211 L.L.C, with particular thanks to K. Ambrose, R. Simone, T. Grundherr, B. Quiambao, and R. Fisher. Technical contributions from Z. Young, R. Vogler, E. Quigley, and A. Trias made this project possible. Funding was provided by the NASA Postdoctoral Program, Bay Area Environmental Research Institute, and Science and Technology Corporation. Funding for instrumentation and aircraft integration is gratefully acknowledged by Ames Research Center Director's funds. Resources supporting

this work were provided by the NASA High-End Computing (HEC) Program through the NASA Advanced Supercomputing (NAS) Division at NASA Ames Research Center.



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





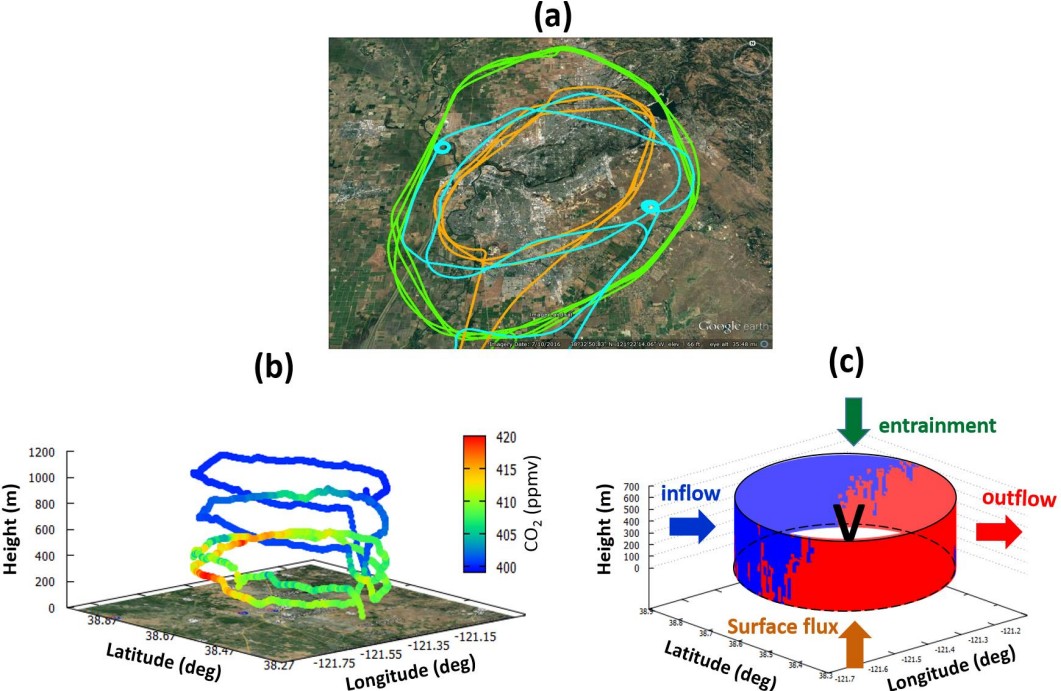

**Figure 1:** (a) Map of AJAX flight tracks on November 18, 2013 (orange), July 29, 2015 (cyan), and November 17, 2015 (green) plotted in Google™ Earth. (b) Vertical measurements of $CO_2$ mixing ratio on November 17, 2015, and (c) simple illustration of airflow [Kg m$^{-2}$ s$^{-1}$] passing through cylinder (over Sacramento). The shading represents the pressure [hPa = Kg m$^{-1}$ s$^{-2}$] divided by the wind vector [m s$^{-1}$] normal to the cylinder. The blue and red represent inflow and outflow, respectively. The vertical mass transfer through the top and bottom are referred to as the entrainment and surface flux, respectively.






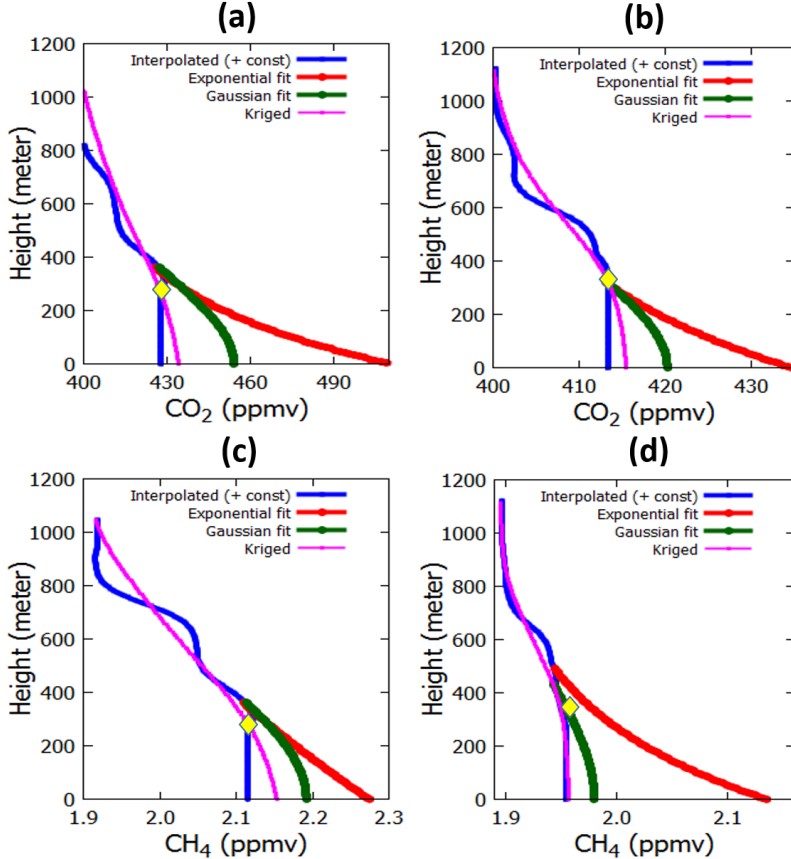

**Figure 2: The CO₂ and CH₄ mixing ratios for (a, c) November 18, 2013, around 38.75º N, 121.27º W and for (b, d) November 17, 2015, around 38.84ºN 121.27º W for CO₂ and 38.58º N, 121.13º W for CH₄. The yellow diamond indicates the altitude of the lowest flight data. The kriged values (magenta), interpolated values with exponential weighting function (blue), and extrapolated values using constant (blue line below diamond), gaussian fit (green), and exponential fit (red) are compared. The empirical fits were generated based on the approach by Gordon et al. (2015).**





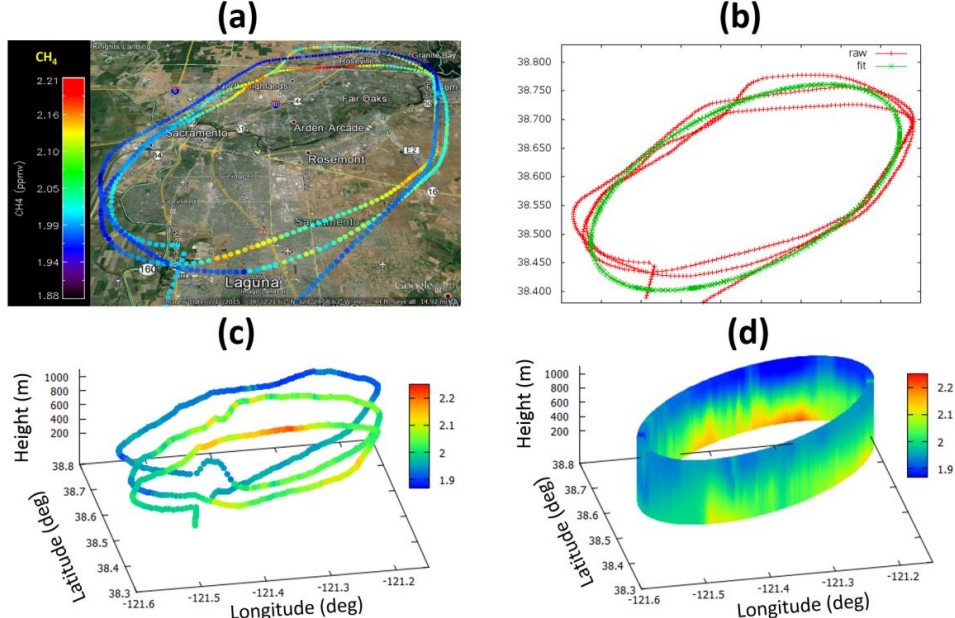

Figure 3: (a) Map of AJAX flight tracks colored by CH₄ mixing ratio for November 18, 2013, plotted in Google™ Earth. (b) The data (red) fitted to an oval (green). The observed CH₄ mixing ratios (c) are kriged to generate the cylindrical surface (d). The axes of the oval are approximately 50 and 40 km.

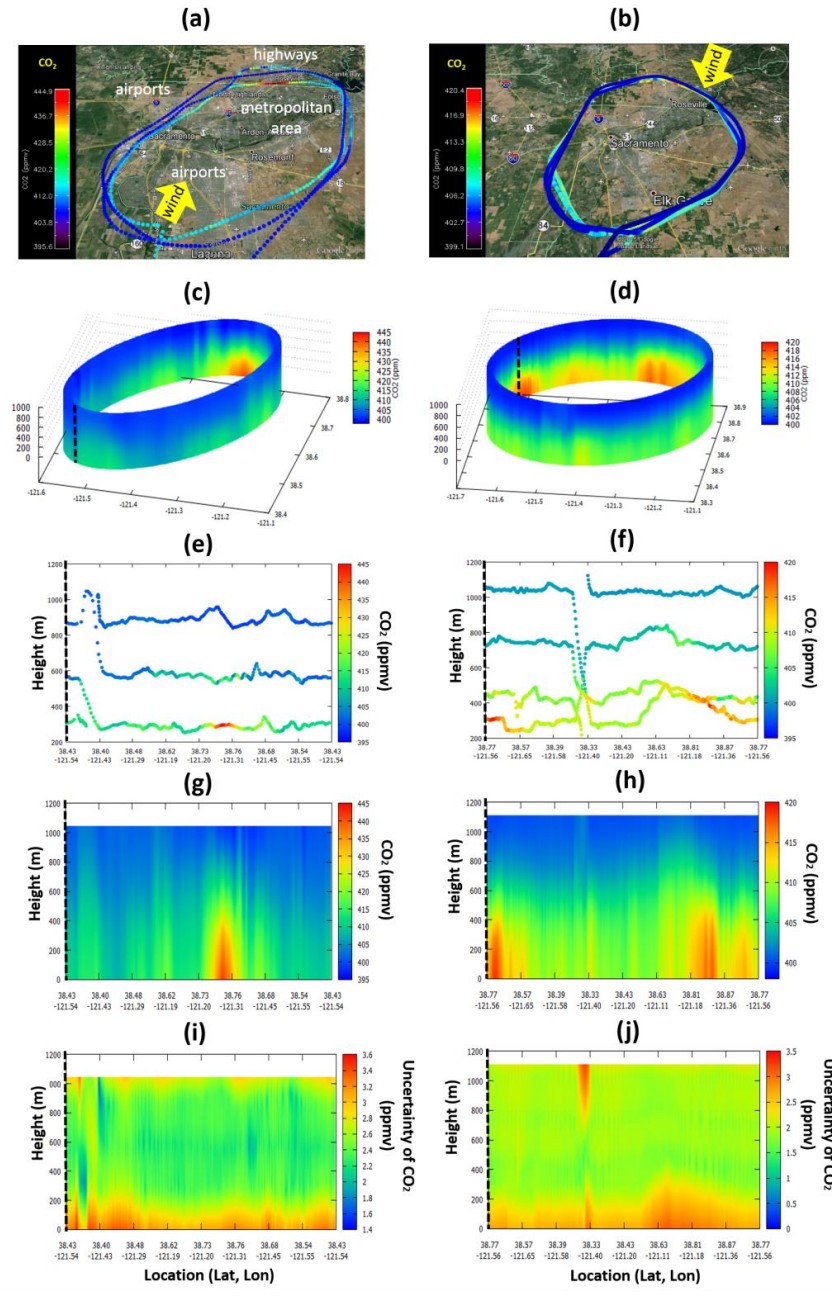


**Figure 4: (a, b, e, f) Measured $CO_2$ mixing ratio, (c, d, g, h) the kriged $CO_2$ mixing ratio, and (i, j) the kriging uncertainty at each grid point on (left) November 18, 2013 and (right) November 17, 2015. The yellow arrows represent the dominant wind directions and the black dashed lines indicate where the surface is split open.**





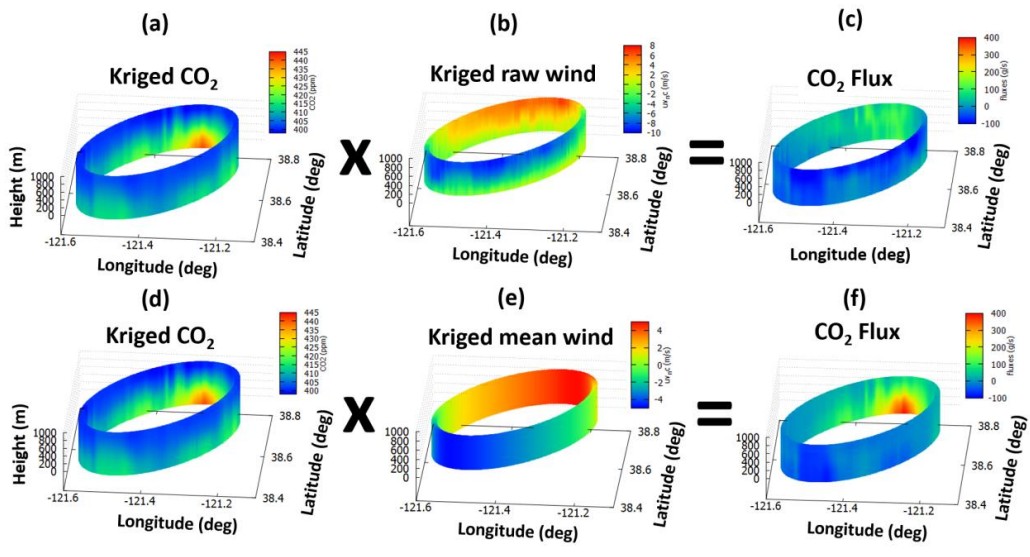

**Figure 5:** (a, d) Kriged $CO_2$ mixing ratio, (b) spatially varying raw measured wind, (e) mass balanced measured wind, (c) $CO_2$ flux using raw measured wind, (f) $CO_2$ flux using mass balanced wind on November 18, 2013. In panels (b, e), the blue color represents the inflow toward (and red outflow from) the cylinder so that it is defined as negative (positive) wind. The background $CO_2$ was chosen as the minimum mixing ratio at each vertical level.





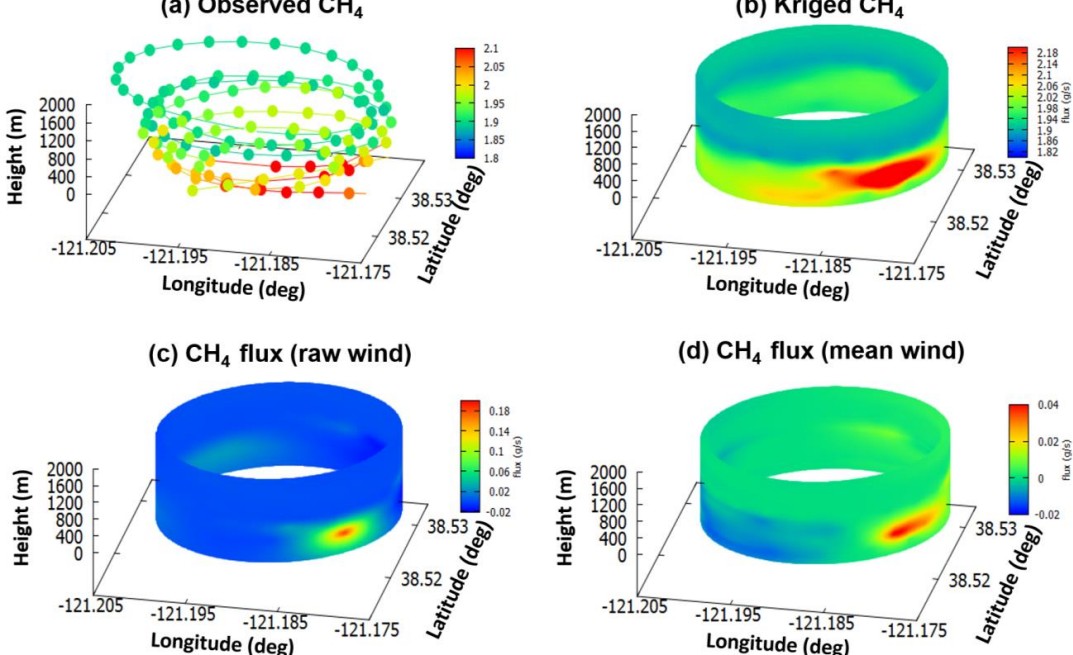

Figure 6: (a) The observed $CH_4$ mixing ratio, (b) the kriged $CH_4$ mixing ratio, (c) the $CH_4$ flux using raw measured wind, (d) the $CH_4$ flux using mass balanced mean wind (for each layer) over the landfill location on July 29, 2015. The fluxes in (c, d) are computed based on equation (2). The background value was chosen as the minimum value at each vertical level. The approximate diameter of the cylinder is 3 km.



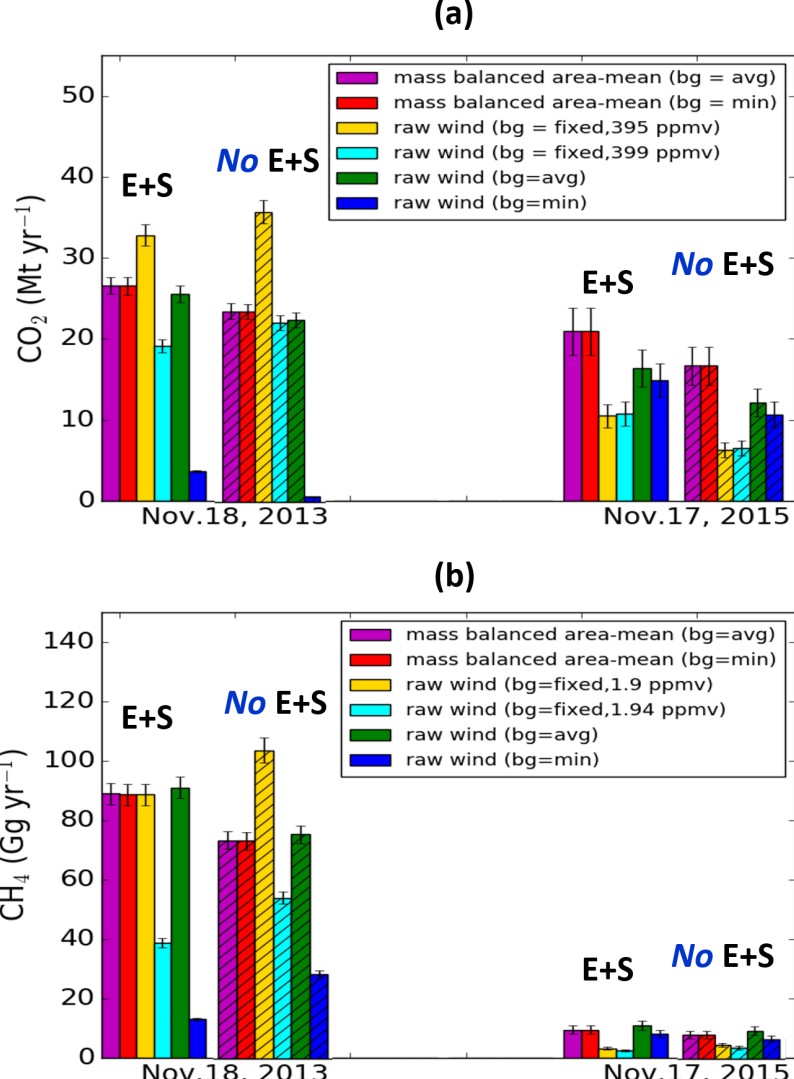

**Figure 7: Urban-scale (a) CO₂ [Mt yr⁻¹] and (b) CH₄ [Gg yr⁻¹] emission rate estimates using raw wind and mass balanced area-mean wind, with different treatments of background values. Solid bars represent emission estimates with entrainment and surface flux (E+S), and hatched bars represent the corresponding emission estimates without consideration of entrainment and surface flux (No E+S). Error bars represent the uncertainty of the total emission fluxes. The average and minimum values for background are computed at each vertical level (about 7 meter interval), and the fixed value alternatives are 395 or 399 ppm for CO₂ and 1.90 or 1.94 ppm for CH₄ for all altitudes.**






## (a)

| AJAX flight date | | Urban Scale (large loop) | | | |
|---|---|---|---|---|---|
| | | November 18, 2013 | | November 17, 2015 | |
| | | CO$_2$ (Mt yr$^{-1}$) | CH$_4$ (Gg yr$^{-1}$) | CO$_2$ (Mt yr$^{-1}$) | CH$_4$ (Gg yr$^{-1}$) |
| Mean wind | Bg= min | 26.55±1.06 | 88.82±3.55 | 20.93±2.93 | 9.48±1.33 |
| Mean wind | Bg =avg | 26.58±1.06 | 88.99±3.56 | 20.92±2.93 | 9.51±1.33 |
| Raw wind | Bg = min | 3.68±0.15 | 13.00±0.52 | 14.92±2.10 | 8.16±1.14 |
| Raw wind | Bg=avg | 25.47±1.02 | 91.52±3.66 | 16.36±2.29 | 10.90±1.53 |



## (b)

| AJAX flight date | | Local Scale (small loop): July 29, 2015 | | | |
|---|---|---|---|---|---|
| | | Landfill | | Rice Field | |
| | | CO$_2$ (Mt yr$^{-1}$) | CH$_4$ (Gg yr$^{-1}$) | CO$_2$ (Mt yr$^{-1}$) | CH$_4$ (Gg yr$^{-1}$) |
| Mean wind | Bg= min | 0.22±0.08 | 7.83±2.74 | 0.25±0.04 | 2.63±0.45 |
| Mean wind | Bg=avg | 0.22±0.08 | 7.67±2.68 | 0.25±0.04 | 2.60±0.44 |
| Raw wind | Bg= min | 0.37±0.13 | 10.26±3.59 | 0.16±0.03 | 1.59±0.27 |
| Raw wind | Bg= avg | 0.22±0.08 | 8.13±2.85 | 0.25±0.04 | 2.66±0.45 |


**Table 1: The calculated CO$_2$ and CH$_4$ emission fluxes using different wind methodology (raw or mean wind) and different background values (minimum or average values at each level) for (a) urban scale on two different flight (November 18, 2013 and November 17, 2015) and (b) local scale on July 29, 2015. The vertical mass transfer (entrainment and surface flux) is taken into account.**
