# Peer review of "Quantification of CO2 and CH4 emissions over Sacramento, California based on divergence theorem using aircraft measurements"

_Atmospheric Measurement Techniques, 2018_

## Referee Comment (RC2)

**Review: Ryoo, AMTD, 2018**

**General Comments:**

This manuscript includes emission estimates of CO2 and CH4 from the Sacramento, California, area from aircraft measurements on three different days. It presents some important and interesting investigations on the sensitivity of flux estimation toward the mass balance method used, the treatment of wind measurements, the choice of background, the inclusion of entrainment at the top of the boundary layer, and different interpolation and extrapolation methods. Still, I think the manuscript needs improvement in the structure, explanation of the methods used as well as the presentation of obtained data and results. The manuscript needs major revisions before it can be accepted for publication in AMT.

The structure of the manuscript could be improved with respect to the different sensitivity studies. I recommend choosing one "best-conduct" approach, explaining and using it for the flights first, and then doing the sensitivity studies and relating their results to this "best-conduct" approach to see each choices influence on the flux calculation individually. Thus there would be one section each on the mass balance method used (Gauss vs. downwind curtain), the treatment of wind measurements (raw winds vs. mass-balanced winds), the choice of background (min vs. avg), the inclusion of entrainment at the top of the boundary layer, and different interpolation (kriging, vertical interpolation, …) and extrapolation methods (kriging, constant, exponential weighting function, Gaussian fit).

The presentation of measured data differs for the three flights and the two compounds CO2 and CH4. Please choose the same set of figures for each flight, making sure all important data used in the flux calculation (like wind speed) is shown for all flights.

The results of using different treatment of input data are the different flux estimates. These numbers are often only named in the text. I think a tabular representation of results for each sensitivity study (or two combined studies, like in Table 1) would increase the readability of the manuscript. I really liked Table 1 and its discussion.

I am not sure if all of your flights are well suited for flux estimation. Generally you would look for a well-mixed boundary layer in order to decrease the uncertainty of flux below the lowest flight height. On November 17, 2015, the winds are quite different from one flight level to another and very weak in the lowest level. Furthermore, detecting the highest CO2 concentration in an upwind part of the flight path shows that this day is not suitable for flux estimation. Also the local flux estimates on July 29, 2015, show low wind speed with changing direction.

Your calculation of mass-balanced wind is interesting. It seems as if the mass-balanced wind in Fig. 5 is constant with height. Shouldn't it vary with height because you use the average wind for each level? What is your surface condition? Why is kriging used? Please explain in more detail how the "mass-balanced" wind field is generated? The difference in the flux estimate does not surprise me (l. 359). You have very good wind measurements, and you definitely see a change in wind speed with height in Fig. 5b. Low wind speed with high concentrations and high wind speed with low

concentration might result in the same flux. Thus, Fig. 5c looks quite logical to me. If you remove all your information on the vertical wind speed change, as you did in Fig. 5e, then of course the flux only represents the concentration measurement. Why do you neglect your information on the wind situation? Did you ever calculate the mass-imbalance of the raw winds? Is it significant?

Your use and understanding of the divergence method seems flawed. Please review the Gauss theorem and describe it correctly in l. 257. I do not believe that a background is necessary for this method. You simply calculate all fluxes through the surface of the cylinder (outflow – inflow) and thus receive the change of mass within the cylinder. Please check the publication of Conley et al. (2017), which nicely explains the application of Gauss theorem on aircraft mass balance flights. For your second method (the curtain downwind of the sources) you definitely need a background and here the influence of choice of the background value on the flux estimate is quite interesting.

Your calculation of the flux through the top of the cylinder is useful, but I do not understand how you determine the surface flux. First, with the Gauss theorem you need to assume that all the mass change inside the cylinder (e.g. what leaves through the surface) comes from the sources on the ground, thus what you determine is the surface flux. Second, on the ground the vertical wind speed is zero. How can the surface flux calculated with your method then be different from zero?

Finally, could you calculate an overall uncertainty of your flux estimates from the different calculation methods and treatment of input parameters?

Please improve the consistency of your terminology. For example you defined the two ways of treating your wind measurements as "raw wind" and "mass-balanced wind". In the following manuscript you then repeatedly use mean wind, measured wind, averaged wind, area-mean wind…

I also recommend grammar checking by an English native speaker and thorough checking of references to Figures and Sections. Furthermore, please increase the size of the axis labels and color bars on most of your plots. They are not readable.

**Specific Comments and Technical Corrections:**

l. 43: Which meteorological factors?

l. 46: "emissions fluxes" should become "emission fluxes". Please check the whole manuscript.

ll. 48-49: This should be reformulated due to the low winds on Nov. 17, 2015, and the high concentrations in the upwind part of the flight pattern.

ll. 49-50: The wind variability and seasonality has not been investigated in this study. Please reformulate.

l. 51: Where do you show the influence of the distance to the emission sources?

l. 58: What is your "modeling strategy"? Do you do any modeling?

l. 61: Why don't you mention your investigation on background and wind treatment in the Abstract?

l. 65: Introduce abbreviations once and then use during remainder of manuscript (e.g. GHG).

l. 66: Is air-quality important in this study?

l. 69: Check your use of "give rise to"…

l. 72: What is the "role of human behavior in altering the emissions"? Do we need to know it for national emission estimates?

l. 76: What are indirect emissions?

l. 78: Please give an example of a bottom-up inventory using proxy data to achieve fine spatial resolution.

ll. 83-86: What about flux estimates of European cities?

l. 94: What do you mean with "those efforts reach general agreement on emission inventories across the cities"?

l. 106: Supplementary Material

l. 108: Do you really mean "uniform vertical mixing", or maybe "uniform distribution of trace gases"?

l. 110: Does this sentence ("These studies…") apply to the first or second category, or maybe both??

l. 112: Does the "single-screen multi-transect method" really depend on constant wind speed? You could also use average wind at each transect or even raw wind at each measurement point.

l. 119: The cylinder pattern should be "around" a source and not only "near".

l. 126: Why are the additional point sources considered sources of uncertainty?

l. 131: I think we all got the concept of three-dimensional space (delete the parentheses).

l. 134: The PBLH is not hard to measure. It is relatively easily determined from a vertical profile of temperature and humidity as you have done in this study. You even stated that the different approaches you used led to similar results. So what is difficult with respect to the PBLH? It is certainly difficult to model correctly, like you stated that substantial differences exist between models and reanalysis data. Also consider the large diurnal variability of PBLH.

l. 135: Please don't use "observed" in connection with models. This might confuse.

l. 139: Do you really think the execution of flights is a goal of the study? Or is it merely necessary for the other goals?

l. 145: Which "value"?

l. 153: Describe the three flights here and mention figure 1. It is not mentioned at all in the text.

l. 154: How many whole-air standards do you use for calibration?

l. 159: Take off time is not sufficient. Please give the total flight times in UTC. Using Pacific Standard Time and Pacific Daylight Time here needs more explanation on why you give take-off times in different ways.

Sect. 2.2: Review order of sections: I would first discuss the interpolation method and then the extrapolation.

l. 163: Reformulate: "Because the lowest flight level was typically between 250 m and 380 m above the surface …"

l. 165: Sentence needs restructuring: The "unmeasured values" lead to uncertainty whether or not a "well-mixed layer assumption" is made. Split sentence!

l.168: Refer to Figure 2 here. There is no reference to it in the text.

l. 169: What do you mean by "elevated" plume? Is it lifted of the ground or are there large enhancements of the concentration?

l. 170: How exactly do you derive the background level? What is the "lowest flight measurement"? Has it got anything to do with the "lowest flight level" which you use in the formula? Is the background only determined from the lowest flight level? Why are you talking about background at this point? It is a section on extrapolation to the ground. Do you also extrapolate the background values? What do X and t stand for?

l. 172: Do you mean that the details of the method are described in Gordon et al. (2015)?

l. 173: How is the Gaussian distribution of the plume dispersion calculated?

Sect. 2.3: Consider renaming the section to "Measurement interpolation".

ll.176-186: Should this be a separate section called: Projection of data to cylinder surface?

l. 182: What is Y?

l. 190: Refer to Fig. S4 as you show these differences there. Consider overplotting the measurements on the kriged and interpolated fields for better assessment of your result that kriging better captures individual plume features. What altitude range do the elliptical cylinder plots cover? Ground to PBLH? Please state in the figure caption.

ll. 214-227: Consider a separate section on uncertainties.

l. 216: Not only downwind interpolated values induce uncertainty. Upwind values as well.

l. 226: Add "observations" behind "direction".

l. 229: Remove "to the choice of background value and" because this is not the topic of this section. You do not investigate the wind characteristics but the treatment of wind measurements.

l. 230: Remove "In one"

l. 231: What are "measured points"? How did you measure them?

l. 233: Stick to one tense (averaged, equaled).

l. 233: "By assuming non-divergence, mass can be balanced." This is correct, but is this really what you need here?

l. 242: Do you assume PBLH to be constant during your flights? At what time during the flight did you measure the profile?

l. 243: Is the boundary layer "growing" during your flights? How do you know?

l. 248: How is $C(t,z)$ determined? How can one point surround the top of the cylinder? How is the background defined here?

l. 252: Is the entrainment calculated from the kriged data?

l. 263: Flux is defined through a surface. Thus it cannot be "inside" the cylinder.

ll. 265-280: See my comment in the General Comments section on the use of a background value with the Gaussian divergence theorem. If you consider inflow and outflow, you do not need a background. In your formula the result should be invariant to the value of background mixing ratio chosen if you consider positive contributions as outflow and negative contributions as inflow.

l. 291: Use present tense.

l. 295: Remove "concentration".

l. 299: How is the kriged estimate less arbitrary in an area far away from measured values? What assumptions is it based on? Is the state of the PBL (stable/unstable) taken into account?

l. 300: You don not mention the Gaussian fit method depicted in Figure 2 at all.

Sect. 3.1: What is the influence of the different choice of interpolation and extrapolation on the flux estimate? Here a table similar to Table one would be great.

l. 304: Remove "gap of the".

ll. 314-320: Please mark all the locations mentioned in the text on a map so the reader can confirm your statement.

l. 325: Present tense.

l. 327: Please check "a farther".

l. 330: Maybe use the last sentence of this paragraph as its first. Good introduction.

l. 350: The PBLH you determine from the vertical profile might have an uncertainty of <1%, but is this value representative for the whole measurement area with this accuracy? What about changes over time and with the location? How does a less defined PBLH influence the uncertainty?

Fig. S6: Looking at your method of estimating PBLH there seems to be a possible error of more than 1 % as well. In Fig. S6d it becomes clear, that you use the 50 m averaged values for checking the gradients. Then you place it at the top of the layer with highest gradient. Here it is visible, that this point is easily 40 m above the layer where a 20 m averaged profile would see the gradient. Thus your uncertainty is around 50 m, which would be almost 10 % for a PBLH of 600 m.

l. 355 ff: See my comment on the treatment of "mass-balanced wind" in the General Comments section.

Sect. 3.3: Please already refer to your Table 1 when naming the results.

l.267: Why is there an "actual" location of the rice field? Pleas show locations on a map rather than just giving coordinates. This is very hard to visualize for a reader.

l. 370: "the local emissions are attributed to these high flux estimates". Did you mean: "The high flux estimates are attributed to the local emissions"?

l. 374: Formulation: "mean wind vector at the dominant wind direction (positive and one direction) and speed". How is this calculated?

l. 381: There is no Table 2.

l. 387: Raw wind is displayed in the bottom two lines.

ll. 390-391: This sentence is incomplete and not logical.

l. 394: Table 1 shows a range of 3.68 - 26.58 Mt CO2 $yr^{-1}$ for the whole city.

l. 396 ff: Here you investigate the difference between using the complete ellipse and only the downwind part. This should be a separate section and the results should be presented in another table.

l. 399: Change "From this study,…" to "According to these calculations…"

l. 401: Table 1 gives a range of 13-92 Mt CO2 $yr^{-1}$ for Nov. 18, 2015.

l. 402: Please indicate "Region-3" on a map.

l. 405: Is vi) the same as i)?

l. 405: Which of these does "This" refer to?

ll. 415-422: "Note … Table 2)." All this is repetition to before and not about the topic of this section which is "vertical mass transfer".

l. 428: Remove "First,"

l. 431: Specify: "different flux calculation methods"

l. 453: There is a contradiction here "the final flux estimates become similar", because the beginning of the sentence states that the background value is a major source of uncertainty.

l. 459: Insert "that" after "suggesting".

ll. 460-468: This section is a general overview of the flight results and should be plasec earlier in the Conclusions.

l. 463: An overview of wind conditions should also be placed in the Results section.

l. 464: This result (isolated high concentrations of CO2) has not been shown in the Results section either.

l.470: Why did you expect sources to be concentrated on the downwind side?

l. 471: "Furthermore" does not fit here.

l. 471: Wind variability definitely influences the flux estimates, not only during different times of the year. So this seems logical. It would be much more interesting how large the uncertainty due to this is assumed to be.

l. 475: The size of the ellipse is another factor that appears here for the first time in the manuscript. There is no data given on how large your ellipses were and what the influence is in the Results section.

l. 480 and 481: Remove two of the three "further".

l. 482: Do you really want to assess: "seasonality of sensitivity of emission estimates"? Just start with seasonality of emissions first.

l. 484: Where do you show the sensitivity of emission estimate uncertainty to temperature and potential temperature?

ll. 490-491: This sentence needs some revision and focus.

*Figures*

Fig. 1: There is no shading visible in Fig. 1c.

Fig. 2: What is the "altitude of lowest flight data"? Please indicate the location of these measurements on a map, giving coordinates is not very helpful.

Fig. 3d: Is the ellipse shown from the ground to the highest flight level or which altitude range?

Fig. 4: Please provide headings with the date of the flight for the left and the right column.

Fig. 5: Why is the mean wind kriged? This has not been mentioned in the text.

Fig. 7: Why is there this large space between the two sets of bars? What is "area-mean"?

Table 1: Tables normally have their description above not below them.

*Supplementary Material:*

There appear to be bits and pieces of text strewn throughout the Supplement. Please give them a heading and a number so it becomes clear where they belong and you then may also refer to them from the main manuscript.

Fig. S1: Figure b color bar label is missing.

l. 7: I am not sure if you can say that emissions are "accumulated" downwind. They are transported downwind, but accumulation would mean that there is very slow wind only.

ll. 11-12: This is not true. With a curtain flight it is also possible to detect emissions from more than one point source within the city, throughout the city and downwind. It gets problematic if there are sources further upwind of the city that get mixed with the city plume and cannot be separated from it.

ll. 15: You mention three types of flight patterns in the main manuscript but only show two of them here.

Fig S2: Reformulate "throughout the altitude". Color bar labels are missing.

l. 25: "accumulated" s.a.

l. 28: Why is air at lower wind speeds less dispersive?

l. 28 ff: Reformulate sentence "Both flights …"

l. 30: Who uses continental scale wind for flux estimates?

Fig. S3: This figure is not mentioned in the main manuscript. Please add flight dates to the left of the plots.

l. 36: Consider "falling". For methane the dashed line is blue. Remove "observation".

Fig. S4: Why are there "boxes" or vertical cuts visible in (d)? Does this have to do with gridding? What is the grid size? Could you plot the measurements on top of the interpolated fields? This way it is easier to assess your statement "kriging reflects the individual plume characteristics better". Could you show the extrapolated fields to the ground as well? Which step is performed first: interpolation or extrapolation? Is this described in the text?

Also: Use the same color bar range for all plots.

l. 57: Don't (b) and (d) also show only the subset of the ellipse? Could you change direction of these plots? Then this arrow would not be necessary.

l. 82: Remove the sentence: "The CH4 enhancement was localized near the landfill." This is obvious.

l. 83: Also remove "…, and we … case." This is also obvious.

Fig. S7: This figure is not mentioned in the main manuscript.

Conley, S., Faloona, I., Mehrotra, S., Suard, M., Lenschow, D. H., Sweeney, C., Herndon, S., Schwietzke, S., Pétron, G., Pifer, J., Kort, E. A., and Schnell, R.: Application of Gauss's theorem to quantify localized surface emissions from airborne measurements of wind and trace gases, Atmos. Meas. Tech., 10, 3345-3358, 10.5194/amt-10-3345-2017, 2017.

---

## Referee Comment (RC1) · Anonymous Referee #1 · 19 Oct 2018

Summary

This paper presents a methodology that estimates CO2 and CH4 fluxes using cylindrical flight patterns combined with kriging and Gauss divergence theorem over Sacramento, California and quantifies the corresponding uncertainty. The study finds that fluxes vary as a function of wind pattern, seasonality, background assumption, flight path, and flux estimation approach by a factor of 1.5 to 8. Total flux estimations using the entire circumference are larger than if just downwind region is used. It is stated in the article that using entire circumference to estimate GHGs fluxes allows for accounting of unknown sources that otherwise could be missed.

General Comments (Major Revisions)

Although the paper does make a lot of important and useful points regarding estimation of GHGs emissions with an aircraft, there are many places that are unclear and need to be elucidated before I can accept this article for publication.

First, I am not exactly clear on how the used methodology is different from a traditional mass balance. I see the explanation, but I am not convinced that it provides any information that is not obtained from the standard method. I would like to see the comparison. Please perform standard mass balance and compare it to your method.

Another issue that I find in this article is that the actual plane data is not carefully presented. First it is important to present data for all of the 3 cases in an equal manner. There are different plots for different days and it becomes confusing. For example, finding exact local time of all the flights is difficult (including just take off time is not enough in this case). This information needs to be easily accessible. I could not locate wind measurements for all of the days. Figure S2 (d) is misleading and actually leads to a flawed assumption regarding steady state for November 17, 2015 (more on that later). The paper needs to be reorganized and improve its clarity of presentation.

The explanation of background is very confusing. Given the method presented, background should be everything that flows into the cylinder. I am not following the justification for different background assumptions (you would not want to pick a minimum value in this case). Also the concept of raw and mass-balanced mean wind needs to be better explained. Why averaging winds horizontally achieves mass balance? And if plume is not well mixed, how can you do that? Plume is transported differently at each level. You cannot just assume that all of the levels move at the same rate.

Specific Comments

Line 170: It says, "The background level is derived from the lowest flight measurement." When using kriging method, do you apply kriging to all of the data including the

background?

Line 191: How do you know that kriging approach captures better plume features? Kriging method interpolates data, meaning that it basically guesses it. It has no knowledge of the actual plume dispersion mechanism. The Figure S4 is misleading as it has different color bar scales for different plots. Please make sure that all of the color bars are the same. In actuality you don't know how the plume is changing below your lowest measurement point. Anything that you assume below that point is highly uncertain. It could be almost constant for all we know. It really depends on the location of source. That's why ideally you want to sample well-mixed layer and not partially mixed layer with an aircraft when estimating flux.

Section 2.4: See the comment about the raw wind vs. mass-balanced mean wind in the general comments section.

Figure 1c: I am confused about the following sentence in the caption, "The shading represents the pressure ... normal to the cylinder." What shading? I am not sure I see any shading. Please explain what do you mean here. Also, here you say that blue is inflow and red is outflow. It seems that everything that is in blue should be a background for everything that is in red assuming steady state. Please comment.

Lines 265-270: I do not understand your choice of background. Given your set up you should be using inflow as background. The definitions you describe here are used in regular mass balance because sometimes there is just not enough sampling, but generally they are flawed. Please explain why you are not using inflow. You need to justify your choices with relevant physical processes.

Another important point that you do not mention is an uptake of $CO_2$ by vegetation. That also can affect background and your fluxes quite a bit. I know it is November in two of your cases, but you need to comment on your assumptions. Your case in July could be more problematic with regard to $CO_2$, although there you concentrate on $CH_4$ so it may not matter as much.

Line 306: How come highways and airports are indicative of CH4 emissions? It is not common for these sources to emit any significant CH4. Please explain.

I think using kriging when you do not understand your sources is a risky endeavor. It is better to solve for everything without kriging first and then see how kriging may affect your results. But in your situation you definitely do not want to trust kriging. Using kriging in regular mass balance is also dangerous if you do not have a good understanding of what you are measuring. Unfortunately it is often used without much thought.

For example, see Figure 6 in Conley et al. (2017), the paper also uses the divergence methodology that you apply here, but they are careful to note that you want an optimal number of loops around your source before you can get a stabilized estimate of emissions. They estimated an optimal number of loops to be about 15 to 25. That is the case because turbulent conditions tend to increase the magnitude of random error. I am afraid your sampling here is just too small for a good application of divergence theorem. It is important to acknowledge it. Solve without kriging and see what you can get.

Figure 2S (b and d): You will have to eliminate November 17, 2015 case from your article. You cannot assume steady state conditions on a day with calm to variable winds near the surface. The wind rose is misleading as you mainly show free tropospheric winds, which should not be used for boundary layer flux calculation. Your boundary layer winds have no consistent direction. The data from a local weather station in Sacramento, CA supports that (and actually if you look carefully at your wind data you will see it too in your Figure). This comes back to the point I made earlier, where you need to show your actual wind data from every case. You cannot just pick and choose what you show. It is no surprise that your flux estimations did not work well on that day. None of the aircraft methods would work on that day. It is very important to have a good forecast before you go and fly a mission of this type. I am not sure who designed this flight and for what, but it does not work here for your purpose. Perhaps you can

find another flight that works better.

Please make sure you address all of my comments.

References

Conley, S., Faloona, I., Mebrotra, S., Suard, M., Lenschow, D.H., Sweeney, C., Herndon, S., Schwietzke, S., Pétron, G., Pifer, J. and Kort, E.A., 2017. Application of Gauss's theorem to quantify localized surface emissions from airborne measurements of wind and trace gases. Atmospheric Measurement Techniques, 10(9).
* * *

---

## Author Comment (AC1) · 20 Dec 2018

Response to Reviewer #1

Thank you very much for your efforts on behalf of our paper titled "Quantification of CO2 and CH4 emissions over Sacramento, California based on divergence theorem using aircraft measurements" submitted by Ju-Mee Ryoo, Laura T. Iraci, Tomoaki Tanaka, Josette E. Marrero, Emma L. Yates, Inez Fung, Anna M. Michalak, Jovan Tadic, Warren Gore, T. Paul Bui, Jonathan M. Dean-Day, Cecilia S. Chang.

We found Reviewer #1's comments very helpful, and they played a large role in improving the quality of this paper. Based on the reviewer's comments, we have reformulated our study in three significant ways (as suggested by both reviewers): we have removed the November 2015 flight from the analysis due to low and variable wind speeds; we have included additional plots to make the measured data more directly available to the reader; and we have reorganized the presentation of the material using the scheme suggested by Reviewer #2 (and guided by the confusion we caused Reviewer #1). Some of the related minor comments became immaterial after these major changes were made, but we have addressed the key critiques and suggestions and will incorporate them in the revised manuscript. Enclosed is a point-by-point response to the review comments. We have shaded with gray highlighting the many useful specific comments that will be implemented in the revised manuscript, once we are asked to prepare it.

The response letter to reviewer #1 is provided in the Supplement.

Thank you again for your support and consideration, and we look forward to hearing a positive decision from you.

Sincerely, Ju-Mee Ryoo and Coauthors

Please also note the supplement to this comment:
https://www.atmos-meas-tech-discuss.net/amt-2018-254/amt-2018-254-AC1-supplement.pdf

———————————————

[Figure]

**Supplement:**

**Response letter to Review #1**

*Responses from the Authors are given in blue italicized text throughout: Thank you for your helpful comments. We addressed them to the best of our understanding and appreciate your guidance on areas where you found our discussion or language unclear. Items such as font size, word choice, and grammar which will be fixed in the revised manuscript are noted with gray highlighting.*

**Summary**

This paper presents a methodology that estimates $CO_2$ and $CH_4$ fluxes using cylindrical flight patterns combined with kriging and Gauss divergence theorem over Sacramento, California and quantifies the corresponding uncertainty. The study finds that fluxes vary as a function of wind pattern, seasonality, background assumption, flight path, and flux estimation approach by a factor of 1.5 to 8. Total flux estimations using the entire circumference are larger than if just downwind region is used. It is stated in the article that using entire circumference to estimate GHGs fluxes allows for accounting of unknown sources that otherwise could be missed.

**General Comments (Major Revisions)**

Although the paper does make a lot of important and useful points regarding estimation of GHGs emissions with an aircraft, there are many places that are unclear and need to be elucidated before I can accept this article for publication.

First, I am not exactly clear on how the used methodology is different from a traditional mass balance. I see the explanation, but I am not convinced that it provides any information that is not obtained from the standard method. I would like to see the comparison. Please perform standard mass balance and compare it to your method.

*From the Authors:  The basic approach we used for estimating $CO_2$ and $CH_4$ fluxes is not different from a traditional mass balance. In the traditional mass balance analysis, the incoming mass should be the same as the outgoing mass passing the X-Y plane of the measurements. So, the air pollutants from the city will be observed on the downwind side, along with the wind passing through the region. Similarly with a cylinder of measurements, the mass coming into the cylinder should be equal to the mass leaving the cylinder to satisfy the mass balance.*

*When we used the raw wind measurement, however, we found that the mass of air mass was not conserved, which means we may not fully apply the mass balance approach. That is why we used the mass-balanced mean wind at each grid level, and the mean wind will distribute the greenhouse gas inside the cylinder. We assume that the well-mixed condition applies.*

*Furthermore, one of the advantages of applying the mass balance approach with an oval (enclosed shape) flight path is that an assignment of the background concentration is not required. Because we calculated the mass of the GHG plume that entered and exited the sample volume, the background effectively cancels out. Sometimes (actually almost always) background is not homogeneous, so there can be large uncertainty associated with calculating the background. There are also challenges in isolating the plume. In our approach, such problems did not exist.*

*Our study is also different from the ususal implementation because we used in situ measured winds, not model data.  Furthermore, we used the mean calculated from measured wind at each level, not just one*

*level, as has been the case in many previous studies (Turnbull et al., 2011; Karion et al. 2013, 2015). The difference we adopted here is that 1) we tested the flux estimation using observed airborne high-resolution raw wind at each measurement point and kriged grid and 2) we tested the flux estimation using mean wind (averaged observed raw wind at each layer). The previous "standard mass balance approach" often adopts the wind data from coarse-resolution model output. The uniqueness that our study provides is that we showed how the final flux could be different depending on the wind treatment.*

Another issue that I find in this article is that the actual plane data is not carefully presented. First, it is important to present data for all of the 3 cases in an equal manner. There are different plots for different days, and it becomes confusing. For example, finding the exact local time of all the flights is difficult (including just take off time is not enough in this case). This information needs to be easily accessible. I could not locate wind measurements for all of the days. Figure S2 (d) is misleading and leads to a flawed assumption regarding steady state for November 17, 2015 (more on that later). The paper needs to be reorganized and improve its clarity of presentation.

*A time series plot of CO₂, CH₄, wind, and altitude for each analysis day is shown below and will be included in the revised Supplemental Information.*
*In addition, we will amend the text in Section 2.1 to focus on the time during which sampling occurred and simplify the way we described the implication of daylight savings time in California (PDT vs. PST). Revised text will read: Sampling occurred 21:10 – 22:00 UTC for the November flight (local standard time is UTC minus 8 h, 13:10 – 14:00 PST) and 20:55 – 21:45 UTC for the June flight. Based on comments from both reviewers, we have removed the November 17, 2015 case. The revised manuscript has changes to figures and the table to reflect this removal.*

[Figure]

[Figure]

The explanation of background is very confusing. Given the method presented, background should be everything that flows into the cylinder. I am not following the justification for different background assumptions (you would not want to pick a minimum value in this case).

*Yes, ideally, the cylindrical flight design removes the need for assigning a background value, as you state. Because we calculated the mass of the GHG that entered and exited the cylinder, we are subtracting all GHG upwind from all GHG downwind. So, the background should cancel out.*

*We tested this expectation by calculating the flux using various wind and background treatments. In the linear curtain approach, the edges of the flight transect outside the plume or measurements made upwind usually provide the background values. In these cases, the background concentration can be non-homogeneous and difficult to specify, so there is large uncertainty associated with setting the background. Because we want to compare as directly as possible our method to the studies already in the literature, we started by assigning the background as the minimum value in the layer (as a parallel case to the "edges" method). Then as a sensitivity test, we explored how the results would or would not change with a different choice of background (average of the layer values). When the winds were used in a way which required mass balance ("mean wind" in Table 1), the choice of background value was shown not to matter (first and second lines of Table 1 and Table 2). The "raw wind" case is discussed in detail in the next item.*

*We hope the new structure of the manuscript will better communicate this approach.*

Also, the concept of raw and mass-balanced mean wind needs to be better explained. Why averaging winds horizontally achieves mass balance? And if plume is not well mixed, how can you do that? Plume is transported differently at each level. You cannot just assume that all of the levels move at the same rate.

*The mass-balanced mean wind is the arithmetic mean of the inflow and outflow raw (measured) wind at each vertical level. Separate calculations were performed to test the sensitivity of the calculated flux to the thickness of the vertical levels. These results will be incorporated into an expanded version of Table 1 in the revised manuscript. This table can be seen below. We will expand it in the revised manuscript, separate it into two tables [Table 1 (urban scale), Table 2 (local scale)], and also include a new Table 3 showing comparison with the Turnbull et al. study and bottom-up inventories.*

*Table 1. Urban scale fluxes over Sacramento on November 18, 2013.*

| Background | Wind | | Urban Scale (large loop) | |
|---|---|---|---|---|
| | | | November 18, 2013 | |
| | | | $CO_2$ | $CH_4$ |
| | | | (Mt yr$^{-1}$) | (Gg yr$^{-1}$) |
| min | Mass-balance | 100 m layer avg | 25.6±2.6 | 87.1±8.7 |
| | | 500 m layer avg | 26.6±2.7 | 88.7±8.9 |
| | | Whole column avg | 26.6±2.7 | 88.7±8.9 |
| avg | Mass-balance | 100m layer avg | 25.6±2.6 | 87.4±8.7 |
| | | 500m layer avg | 26.6±2.7 | 89.0±8.9 |
| | | Whole column avg | 26..6±2.7 | 89.0±8.9 |
| min | Raw | | 3.7±0.4 | 13.0±1.3 |
| avg | Raw | | 25.5±2.6 | 91.1±9.1 |

*Table 2. Local scale fluxes over landfill and rice field over Sacramento on July 29, 2015.*

| Background | Wind | | Local Scale (small loop): July 29, 2015 | | | |
| --- | --- | --- | --- | --- | --- | --- |
| | | | Landfill | | Rice Field | |
| | | | $CO_2$ (Mt yr$^{-1}$) | $CH_4$ (Gg yr$^{-1}$) | $CO_2$ (Mt yr$^{-1}$) | $CH_4$ (Gg yr$^{-1}$) |
| min | Mass-balance | 100 m layer avg | 0.2±0.1 | 7.1±2.5 | 0.3±0.04 | 2.6±0.4 |
| | | 500 m layer avg | 0.2±0.1 | 7.1±2.5 | 0.3±0.04 | 2.6±0.4 |
| | | Whole column avg | 0.2±0.1 | 6.9±2.4 | 0.3±0.04 | 2.6±0.5 |
| avg | Mass-balance | 100 m layer avg | 0.2±0.1 | 7.1±2.5 | 0.2±0.04 | 2.6±0.4 |
| | | 500 m layer avg | 0.2±0.1 | 7.1±2.5 | 0.2±0.04 | 2.6±0.4 |
| | | Whole column avg | 0.2±0.1 | 6.9±2.4 | 0.2±0.04 | 2.6±0.4 |
| min | Raw | | 0.4±0.1 | 9.0±3.1 | 0.2±0.03 | 1.7±0.3 |
| avg | Raw | | 0.2±0.1 | 7.1±2.5 | 0.3±0.04 | 2.6±0.5 |

*Table 3. Flux estimates for the Sacramento urban area from measurements made on November 18, 2013.*

| | | CO2 (Mt yr$^{-1}$) | CH4 (Gg yr$^{-1}$) |
| --- | --- | --- | --- |
| Whole cylinder–AJAX | (bg = min, 100m layer avg) | 25.6 ± 2.6 | 87.1 ± 8.7 |
| | (bg = avg, 100m layer avg) | 25.6 ± 2.6 | 87.4 ± 8.7 |
| Curtain –AJAX | (bg = min) | 17.3 ± 1.7 | 64.4 ± 6.4 |
| | (bg = avg) | 8.9 ± 0.9 | 24.1 ± 2.4 |
| Turnbull et al. (2011) | | 13.6 (with uncertainty of ~ 100%) | |
| Vulcan estimates for Sacramento | | 11.7 | |
| CEPAM estimate for Sacramento | | 10.3 | |

*[a] Turnbull et al. (2011) data was collected in 2009; the value given here was converted from the mean reported value of 3.5 Mt C yr$^{-1}$ with a 1.1% yr$^{-1}$ increase in $CO_2$ flux to adjust to 2015.*
*[b] Bottom-up inventory estimates of the annual total emissions from Sacramento County from Vulcan (Gurney et al., 2009) and the California Air Resources Board CEPAM database (Turnbull et al, 2011) are included for comparison. The Vulcan inventory is available only for 2002, and the CEPAM database is available for 2004. We applied a 1.1% yr$^{-1}$ increase in $CO_2$ flux to adjust to 2015.*

*Previous studies used a single coarse resolution model 'mean' wind throughout all altitudes below the PBLH (Turnbull et al., 2011; Karion et al. 2015), and we agree with you that assuming the winds at all levels move at the same rate is not ideal. Therefore, we used our in situ wind measurements to test the impact of this assumption. We calculated the fluxes using wind averaged on each vertical layer (three separate tests with layer thicknesses of 100, 200, and 500 m) and found the results were actually not very much different from the flux estimate found when we use a single, whole column average.*

**Specific Comments**

Line 170: It says, "The background level is derived from the lowest flight measurement." When using the kriging method, do you apply kriging to all of the data including the background?

*We apologize for the wording in the original manuscript at line 170. Our description of the "constant" method was unnecessarily confusing. The blue curves in Figure 2 show what we were trying to describe: at all altitudes below the yellow diamond (lowest flight level), the mixing ratio was presumed to be exactly equal to the value measured at the lowest flight level. Please also see the new figure below. The bottom left panel shows more clearly how the values measured along the lowest flight level are assigned to all grid cells below the measurement altitude. The dashed lines represent an approximate lowest flight level.*

[Figure]

Line 191: How do you know that kriging approach captures better plume features? Kriging method interpolates data, meaning that it basically guesses it. It has no knowledge of the actual plume dispersion mechanism. The Figure S4 is misleading as it has different color bar scales for different plots. Please make sure that all of the color bars are the same. In actuality, you don't know how the plume is changing below your lowest measurement point. Anything that you assume below that point is highly uncertain. It could be almost constant for all we know. It really depends on the location of the source.

That's why ideally you want to sample the well-mixed layer and not partially mixed layer with an aircraft when estimating flux.

*That is exactly why we evaluated several common methods of extrapolation, as shown in Figure 2 and the new figure above. As you point out, we do not know which method will better capture the plume features, especially below the lowest flight level. Because kriging uses characteristics of the measured data to make the interpolation, we expect it to be less arbitrary than some of the other methods, and this is what is seen in figure S4. The measured plume shape (panel b) is captured more faithfully by the kriging method (panel c) than by the exponential method (panel d), as is the subtle variation of mixing ratio around the oval at a given altitude.*

*We agree with you that plume behavior below our lowest measurement point is highly uncertain. We don't think one pattern is correct while others are not. We just suggest one estimate might be better based on the other characteristics that we've examined. Your comments on why a well-mixed layer condition is important for estimating flux are also very true.*

*More attention will be paid to the color bar scales in the revised manuscript, and where appropriate they will have the same scale, such as panels b, c, and d of Figure S4. However, if forcing different flight days to use the same scale makes it more difficult to discern features of interest, then we will not force one day's maximum value to set the scale bar for another day's flight. If different scales are necessary for clarity, we will note the change in the relevant figure caption(s).*

Section 2.4: See the comment about the raw wind vs. mass-balanced mean wind in the general comments section.

*Please see reply above.*

Figure 1c: I am confused about the following sentence in the caption, "The shading represents the pressure . . . normal to the cylinder." What shading? I am not sure I see any shading. Please explain what do you mean here. Also, here you say that blue is inflow and red is an outflow. It seems that everything that is in blue should be a background for everything that is in red assuming steady state. Please comment.

*We will change the word choice in the revised manuscript. "Shading" was intended to mean "colors" of the cylinder. This is just the sign of the air mass flux [kg m$^{-2}$ s$^{-1}$], which is obtained from density multiplied by the wind vector (or pressure divided by the wind vector). Yes, everything in blue represents the inflow air mass, which has negative wind direction, while everything in red represents the outflow air mass, which has positive wind direction.*

Lines 265-270: I do not understand your choice of background. Given your set up you should be using inflow as background. The definitions you describe here are used in regular mass balance because sometimes there is just not enough sampling, but generally, they are flawed. Please explain why you are not using inflow. You need to justify your choices with relevant physical processes.

*We agree with you that we do not need to know the background value for estimating flux when adopting the circular pattern of flight. This is clearly demonstrated in our experiments, showing that the flux estimates were not sensitive to the choice of background (minimum or averaged value). We will work harder on the language in the revised manuscript.*

Another important point that you do not mention is an uptake of $CO_2$ by vegetation. That also can affect background and your fluxes quite a bit. I know it is November in two of your cases, but you need to comment on your assumptions. Your case in July could be more problematic with respect to $CO_2$, although there you concentrate on $CH_4$ so it may not matter as much.

*We agree that considering an uptake of $CO_2$ by vegetation could be an important factor on total flux estimate. When we took a look at CarbonTracker data, we confirmed that there is some contribution of the vegetation of $CO_2$ to the total fluxes in November (for example, in places like Salt Lake City). However, it is hard to consider the biological impact on $CO_2$ flux unless we downscale the model data to the small scale we are interested in. This would be better considered in a further study to completely characterize each sector (biological (vegetation or dairy farm) or anthropogenic (industry)). However, we agree with the reviewer that we should mention the potential problem in the interpretation of the flux estimate we obtained in this study and will include a comment in the revised manuscript.*

Line 306: How come highways and airports are indicative of $CH_4$ emissions? It is not common for these sources to emit any significant $CH_4$. Please explain.

*Dairy farms and landfills are well known sources of $CH_4$. However, one of biggest concerns regarding CH4 emission is the contribution from unknown sources. We see the slight increase of $CH_4$ emission over those sites. Broken pipe lines or other facilities at the airport could be possible sources of $CH_4$, so we called it to attention. But, as you pointed out, this cannot be a deterministic source of $CH_4$. We will modify/rephrase the sentences in the revised manuscript.*

I think using kriging when you do not understand your sources is a risky endeavor. It is better to solve for everything without kriging first and then see how kriging may affect your results. But in your situation, you definitely do not want to trust kriging. Using kriging in regular mass balance is also dangerous if you do not have a good understanding of what you are measuring. Unfortunately it is often used without much thought. For example, see Figure 6 in Conley et al. (2017), the paper also uses the divergence methodology that you apply here, but they are careful to note that you want an optimal number of loops around your source before you can get a stabilized estimate of emissions. They estimated an optimal number of loops to be about 15 to 25. That is the case because turbulent conditions tend to increase the magnitude of the random error. I am afraid your sampling here is just too small for a good application of divergence theorem. It is important to acknowledge it. Solve without kriging and see what you can get.

*We understand your concerns, and we appreciate the "riskiness". When we calculate fluxes based only on the measured data, without filling in the gaps between flight levels, the total flux estimate will obviously be much smaller than when we account for the entire surface of the cylinder using interpolated data. With an urban-scale cylinder (with a circumference on the order of 100 km), it is impossible to map out the entire surface ($\sim$100 $km^2$) with dense measurements. Although kriging cannot be better than actual observations, it can be a good alternative to "mimic" actual data. We disagree with the reviewer's opinion that we solely rely on the kriging without an understanding of the data. We carefully performed the variogram analysis, and carefully chose the kriging parameters (sill, range, and nugget) based on the experimental and theoretical variogram obtained from the actual data we measured.*

Figure 2S (b and d): You will have to eliminate November 17, 2015 case from your article. You cannot assume steady state conditions on a day with calm to variable winds near the surface. The wind rose is misleading as you mainly show free tropospheric winds, which should not be used for boundary layer flux calculation. Your boundary layer winds have no consistent direction. The data from a local weather station in Sacramento, CA supports that (and actually if you look carefully at your wind data you will see it too in your Figure). This comes back to the point I made earlier, where you need to show your actual wind data from every case. You cannot just pick and choose what you show. It is no surprise that your flux estimations did not work well on that day. None of the aircraft methods would work on that day. It is very important to have a good forecast before you go and fly a mission of this type. I am not sure who designed this flight and for what, but it does not work here for your purpose. Perhaps you can find another flight that works better.

*Done. Please see discussion above.*

*Reference*

*Conley, S., I. Faloona, S. Mehrotra, M. Suard, D. H. Lenschow, C. Sweeney, S. Herndon, S. Schwietzke, G Petron, J. Pifer, E. A. Kort, and R. Schnell: Application of Gauss's theorem to quantify localized surface emissions from airborne measurements of wind and trace gases, Atmos. Meas. Tech., 10, 3345-3358, https://doi.org/10.5194/amt-10-3345-2017, 2017.*

*Karion, A. et al.: Methane emissions estimate from airborne measurements over a western United States natural gas field, Geophys. Res. Lett., Vol. 40, 1-5, doi:10.1002/grl.50811, 2013.*

*Karion, A. et al.: Aircraft-Based Estimate of Total Methane Emissions from the Barnett Shale Region, Environ. Sci. Technol. 2015, 49, 8124-8131, DOI: 10.1021/acs.est.5b00217, 2015.*

*Turnbull, J. C., Karion, A., Fischer, M. L., Faloona, I., Guilderson, T., Lehman, S. J., Miller, B.R., Miller, J. B., Montzka, S., Sherwood, T., Saripalli, S., Sweeney, C., and Tan, P.P.:: Assessment of fossil fuel carbon dioxide and other anthropogenic trace gas emissions from airborne measurements over Sacramento, California in spring 2009. Atmos. Chem. Phys., 11, 705–721, 2011, doi:10.5194/acp-11-705-2011, 2011.*

---

## Author Comment (AC2) · 20 Dec 2018

Response to Reviewer #2:

Thank you very much for your efforts on behalf of our paper titled "Quantification of CO2 and CH4 emissions over Sacramento, California based on divergence theorem using aircraft measurements" submitted by Ju-Mee Ryoo, Laura T. Iraci, Tomoaki Tanaka, Josette E. Marrero, Emma L. Yates, Inez Fung, Anna M. Michalak, Jovan Tadic, Warren Gore, T. Paul Bui, Jonathan M. Dean-Day, Cecilia S. Chang.

We found Reviewer #2's comments very useful, and they contribute to improving the

quality of this paper. Based on the reviewer's comments, we have reformulated our study in three significant ways (as suggested by both reviewers): i) we have removed the November 2015 flight from the analysis due to low and variable wind speeds; ii) we have included additional plots to make the measured data more directly available to the reader; and iii) we have reorganized the presentation of the material using the scheme suggested by Reviewer #2 (and guided by the confusion we caused Reviewer #1). We have addressed the primary suggestion and the related minor comments and will incorporate them in the revised manuscript.

Enclosed is a point-by-point response to reviewer #2's comments and is provided in the Supplement. We have shaded with gray highlighting the many useful specific comments that will be implemented in the revised manuscript, once we are asked to prepare it.

Thank you again for your support and consideration, and we look forward to hearing a positive decision from you.

Sincerely, Ju-Mee Ryoo and Coauthors

Please also note the supplement to this comment:
https://www.atmos-meas-tech-discuss.net/amt-2018-254/amt-2018-254-AC2-supplement.pdf

―――――――――――――――――

**Supplement:**

**Response letter to Review #2**

*Responses from the Authors are given in blue italicized text throughout: Thank you for your very helpful feedback. Specific items of language, font size, etc. which will be addressed in the revised manuscript are colored here with gray highlighting.*

**General Comments:**

This manuscript includes emission estimates of $CO_2$ and $CH_4$ from the Sacramento, California, area from aircraft measurements on three different days. It presents some important and interesting investigations on the sensitivity of flux estimation toward the mass balance method used, the treatment of wind measurements, the choice of background, the inclusion of entrainment at the top of the boundary layer, and different interpolation and extrapolation methods. Still, I think the manuscript needs improvement in the structure, explanation of the methods used as well as the presentation of obtained data and results. The manuscript needs major revisions before it can be accepted for publication in AMT.

The structure of the manuscript could be improved with respect to the different sensitivity studies. I recommend choosing one "best-conduct" approach, explaining and using it for the flights first, and then doing the sensitivity studies and relating their results to this "best-conduct" approach to see each choices influence on the flux calculation individually. Thus there would be one section each on the mass balance method used (Gauss vs. downwind curtain), the treatment of wind measurements (raw winds vs. mass-balanced winds), the choice of background (minimum vs. average), the inclusion of entrainment at the top of the boundary layer, and different interpolation (kriging, vertical interpolation) and extrapolation methods (kriging, constant, exponential weighting function, Gaussian fit).

*The structure of the manuscript will be completely reorganized based on your suggestion of a "best conduct" approach followed by variants. More calculations were performed to make our points clear in the revised version of manuscript.*

*The outline of the revised manuscript will be this:*

*1. Introduction*
*2. Data and Methods*
  *2.1 Data Collection*
  *2.2 Data Gridding*
      *2.2.1 Extrapolation to the Surface*
      *2.2.2 Elliptical Fit and Measurement Interpolation (Kriging Method)*
      *2.2.3 Kriged GHG Mixing Ratios*
*3. Flux calculations*
   *3.1 "Best Conduct" (or "base case")*
      *3.1.1 Wind Treatment*
      *3.1.2 Background Mixing Ratios*
      *3.1.3 Planetary Boundary Layer Height (PBLH) and Entrainment*
      *3.1.4 "Best Conduct" calculated fluxes*
   *3.2 Sensitivity Tests*

The presentation of measured data differs for the three flights and the two compounds $CO_2$ and $CH_4$. Please choose the same set of figures for each flight, making sure all important data used in the flux calculation (like wind speed) is shown for all flights.

*We appreciate this comment. We will include the wind data in the revised manuscript (see above) and will revise our figures.*

The results of using different treatment of input data are the different flux estimates. These numbers are often only named in the text. I think a tabular representation of results for each sensitivity study (or two combined studies, like in Table 1) would increase the readability of the manuscript. I really liked Table 1 and its discussion.

*We are glad that Table 1 was clear and helpful. We will expand it in the revised manuscript, separate it into two tables [Table 1 (urban scale), Table 2 (local scale)], and also include a new Table 3 showing comparison with the Turnbull et al. study and bottom-up inventories.*

*Table 1. Urban scale fluxes over Sacramento on November 18, 2013.*

| Background | Wind | | Urban Scale (large loop) | |
|---|---|---|---|---|
| | | | November 18, 2013 | |
| | | | CO$_2$ (Mt yr$^{-1}$) | CH$_4$ (Gg yr$^{-1}$) |
| min | Mass-balance | 100 m layer avg | 25.6±2.6 | 87.1±8.7 |
| | | 500 m layer avg | 26.6±2.7 | 88.7±8.9 |
| | | Whole column avg | 26.6±2.7 | 88.7±8.9 |
| avg | Mass-balance | 100m layer avg | 25.6±2.6 | 87.4±8.7 |
| | | 500m layer avg | 26.6±2.7 | 89.0±8.9 |
| | | Whole column avg | 26..6±2.7 | 89.0±8.9 |
| min | Raw | | 3.7±0.4 | 13.0±1.3 |
| avg | Raw | | 25.5±2.6 | 91.1±9.1 |

*Table 2. Local scale fluxes over landfill and rice field over Sacramento on July 29, 2015.*

| Background | Wind | | Local Scale (small loop): July 29, 2015 | | | |
|---|---|---|---|---|---|---|
| | | | Landfill | | Rice Field | |
| | | | $CO_2$ (Mt yr$^{-1}$) | $CH_4$ (Gg yr$^{-1}$) | $CO_2$ (Mt yr$^{-1}$) | $CH_4$ (Gg yr$^{-1}$) |
| min | Mass - balance | 100 m layer avg | 0.2±0.1 | 7.1±2.5 | 0.3±0.04 | 2.6±0.4 |
| | | 500 m layer avg | 0.2±0.1 | 7.1±2.5 | 0.3±0.04 | 2.6±0.4 |
| | | Whole column avg | 0.2±0.1 | 6.9±2.4 | 0.3±0.04 | 2.6±0.5 |
| avg | Mass-balance | 100 m layer avg | 0.2±0.1 | 7.1±2.5 | 0.2±0.04 | 2.6±0.4 |
| | | 500 m layer avg | 0.2±0.1 | 7.1±2.5 | 0.2±0.04 | 2.6±0.4 |
| | | Whole column avg | 0.2±0.1 | 6.9±2.4 | 0.2±0.04 | 2.6±0.4 |
| min | Raw | | 0.4±0.1 | 9.0±3.1 | 0.2±0.03 | 1.7±0.3 |
| avg | Raw | | 0.2±0.1 | 7.1±2.5 | 0.3±0.04 | 2.6±0.5 |

*Table 3. Flux estimates for the Sacramento urban area from measurements made on November 18, 2013.*

| | | CO2 (Mt yr$^{-1}$) | CH4 (Gg yr$^{-1}$) |
|---|---|---|---|
| Whole cylinder–AJAX | (bg = min, 100m layer avg) | 25.6 ± 2.6 | 87.1 ± 8.7 |
| | (bg = avg, 100m layer avg) | 25.6 ± 2.6 | 87.4 ± 8.7 |
| Curtain –AJAX | (bg = min) | 17.3 ± 1.7 | 64.4 ± 6.4 |
| | (bg = avg) | 8.9 ± 0.9 | 24.1 ± 2.4 |
| Turnbull et al. (2011) | | 13.6 (with uncertainty of ~ 100%) | |
| Vulcan estimates for Sacramento | | 11.7 | |
| CEPAM estimate for Sacramento | | 10.3 | |

*[a] Turnbull et al. (2011) data was collected in 2009; the value given here was converted from the mean reported value of 3.5 Mt C yr-1 with a 1.1% yr$^{-1}$ increase in $CO_2$ flux to adjust to 2015.*
*[b] Bottom-up inventory estimates of the annual total emissions from Sacramento County from Vulcan (Gurney et al., 2009) and the California Air Resources Board CEPAM database (Turnbull et al, 2011) are included for comparison. The Vulcan inventory is available only for 2002, and the CEPAM database is available for 2004. We applied a 1.1% yr$^{-1}$ increase in $CO_2$ flux to adjust to 2015.*

I am not sure if all of your flights are well suited for flux estimation. Generally, you would look for a well-mixed boundary layer in order to decrease the uncertainty of flux below the lowest flight height. On November 17, 2015, the winds are quite different from one flight level to another and very weak at the lowest level. Furthermore, detecting the highest $CO_2$ concentration in an upwind part of the flight path

shows that this day is not suitable for flux estimation. Also the local flux estimates on July 29, 2015, show low wind speed with changing direction.

*Based on the comments of both reviewers, we have decided to remove the November 2015 case due to the vertical variability in wind speeds. However, we still believe July 29, 2015 is appropriate for testing the "closed shape" approach for flux estimates and have retained these cases in the current analysis.*

Your calculation of mass-balanced wind is interesting. It seems as if the mass-balanced wind in Fig. 5 is constant with height. Shouldn't it vary with height because you use the average wind for each level? What is your surface condition? Why is kriging used? Please explain in more detail how the "mass-balanced" wind field is generated? The difference in the flux estimate does not surprise me (l.359). You have very good wind measurements, and you definitely see a change in wind speed with height in Fig. 5b. Low wind speed with high concentrations and high wind speed with low concentration might result in the same flux. Thus, Fig. 5c looks quite logical to me. If you remove all your information on the vertical wind speed change, as you did in Fig. 5e, then, of course, the flux only represents the concentration measurement. Why do you neglect your information on the wind situation? Did you ever calculate the mass-imbalance of the raw winds? Is it significant?

*Thank you for your insightful discussion. Yes, the mass-balanced wind should vary with height when we use the average wind for each level. We have tested a variety of thicknesses for the vertical levels, and the results are now reported in the modified Table 1 (above). The original Fig. 5e was showing the "whole column average" case, and we apologize for not specifying this clearly in the original text. For reference, we show here the wind field with all four treatments. Panel (d) here is the same as Figure 5e in the original manuscript. We intend to include panel (b) here in an extended version of Fig. 5 in the revised manuscript.*

[Figure]

**Wind treatment on November 18, 2013**

(a) Raw wind

(b) 100m vertically averaged wind

(c) 500m vertically averaged wind

(d) Whole column vertically averaged wind

Your use and understanding of the divergence method seems flawed. Please review the Gauss theorem and describe it correctly in l. 257. I do not believe that background is necessary for this method. You simply calculate all fluxes through the surface of the cylinder (outflow – inflow) and thus receive the change of mass within the cylinder.

*We do not need to know the background value for estimating flux when adopting the circular pattern of flight. This is clearly demonstrated in our experiments, showing that the flux estimates were not sensitive to the choice of background (minimum or averaged value). We will work harder on the language in the revised manuscript.*

Please check the publication of Conley et al. (2017), which nicely explains the application of Gauss theorem on aircraft mass balance flights. For your second method (the curtain downwind of the sources) you definitely need a background, and here the influence of choice of the background value on the flux estimate is quite interesting.

*Thank you for the suggestion. We referenced Conley et al. (2017), and looked at their approach. We carefully considered their method and compared them to our approach.*

Your calculation of the flux through the top of the cylinder is useful, but I do not understand how you determine the surface flux. First, with the Gauss theorem you need to assume that all the mass change inside the cylinder (e.g. what leaves through the surface) comes from the sources on the ground, thus what you determine is the surface flux. Second, on the ground the vertical wind speed is zero. How can the surface flux calculated with your method then be different from zero?

*Since the lowest flight altitude is about ~250 meters above the ground level, we can't directly measure the surface $CO_2$ or $CH_4$ concentration or wind. The vertical velocity near the surface was very small, but it was not zero (W = -0.006 m/s for November 18, 2013 case). So we could still calculate the surface flux part. But the surface flux was much smaller than the entrainment flux, and the entrainment flux itself was not much compared to the flux computed on the "cylindrical" (wall) surface. Although the contribution of surface flux uncertainty to the total flux estimate uncertainty could be improved, we decided not to focus on this in the current study since the contribution of vertical mass transfer to the total flux estimate is relatively small.*

Finally, could you calculate an overall uncertainty of your flux estimates from the different calculation methods and treatment of input parameters?

*We have determined the overall uncertainty of our flux estimates based on the kriged and mass balance wind methods. By assuming that the errors of each factor are Gaussian in nature, and each measurement (e.g., $CO_2$ and wind) is independent (no covariance), we estimate the overall uncertainties in the calculated flux by calculate the relative uncertainties from each point to the adding the fractional uncertainties of the kriged $CO_2$, $CH_4$, and winds in quadrature, as in Nathan et al. (2015). We also consider the uncertainty due to estimate of PBLH. We will add a brief discussion to the revised manuscript comparing the distribution of the calculated fluxes reported in the revised Tables to the calculated 10% uncertainty estimate.*

Please improve the consistency of your terminology. For example, you defined the two ways of treating your wind measurements as "raw wind" and "mass-balanced wind". In the following manuscript you then repeatedly use mean wind, measured wind, averaged wind, area-mean wind…
*Very important points. Thank you for providing "fresh eyes" to notice this with. We should stick to using "raw wind" and "mass-balanced wind". We will improve this in the revised manuscript.*

I also recommend grammar checking by an English native speaker and thorough checking of references to Figures and Sections. Furthermore, please increase the size of the axis labels and color bars on most of your plots. They are not readable.
*We will make sure to improve the size and the quality of the plots in the revised manuscript.*

**Specific Comments and Technical Corrections:**
l. 43: Which meteorological factors?
*We meant the wind speed and measurement. We will rephrase it in the revised manuscript.*

l. 46: "emissions fluxes" should become "emission fluxes". Please check the whole manuscript.
*Thanks. We will double check this in the revised manuscript.*

ll. 48-49: This should be reformulated due to the low winds on Nov. 17, 2015, and the high concentrations in the upwind part of the flight pattern.
*We agree with you and have decided that this flight was not adequate for flux estimates. Hence, we will remove this flight from our analysis.*

ll. 49-50: The wind variability and seasonality has not been investigated in this study. Please reformulate.
*We agree with you. In the revised manuscript, we will remove those parts.*

l. 51: Where do you show the influence of the distance to the emission sources?
*We don't show this. We will correct the sentences in the revised manuscript.*

l. 58: What is your "modeling strategy"? Do you do any modeling?
*Here we meant the "statistical modeling". Using one of the geostatistical methods, kriging, we filled the gaps between measured data.*

l. 61: Why don't you mention your investigation of the background and wind treatment in the Abstract?
*This is a good idea. In the revised manuscript, we will mention our background and wind treatment in the abstract.*

l. 65: Introduce abbreviations once and then use during the remainder of the manuscript (e.g., GHG).
*Yes, we will do that in the revised manuscript.*

l. 66: Is air-quality important in this study?
*No, but accurate emission estimates will affect "air quality" and its regulation, so this is why we mentioned this in the introduction.*

l. 69: Check your use of "give rise to"…
*We will change the word to "causes".*

l. 72: What is the "role of human behavior in altering the emissions"? Do we need to know it for national emission estimates?
*Good points, we don't need to do know, so we will remove this part.*

l. 76: What are indirect emissions?
*"Indirect emission" was intended to describe information in emission databases which are inferred or extrapolated or given a time dependence that is not directly measured. We will find a more direct way to state this in the revised manuscript.*

l. 78: Please give an example of a bottom-up inventory using proxy data to achieve fine spatial resolution.
*Vulcan inventory $CO_2$ data. (Gurney et al., 2009)*

ll. 83-86: What about flux estimates of European cities?
*This a great point. So we will add studies for European cities in the revised manuscript:* (Peylin et al, 2005; Kountouris et al., 2018).

l. 94: What do you mean with "those efforts reach general agreement on emission inventories across the cities"?
*We will rephrase It in the revised manuscript. "While those efforts reach general agreement on emission inventories across the cities" will be changed to "Since current emission inventories do not consider individual characteristics of each city, they have limitations…".*

l. 106: Supplementary Material
*We will reword it.*

l. 108: Do you really mean "uniform vertical mixing", or maybe "uniform distribution of trace gases"?
*We will reword it. "uniform vertical mixing" will be changed to "uniform distribution of trace gases with altitude within the PBL and with time".*

l. 110: Does this sentence ("These studies…") apply to the first or second category, or maybe both??
*These studies apply to the first category.*

l. 112: Does the "single-screen multi-transect method" really depend on constant wind speed? *You could also use average wind at each transect or even raw wind at each measurement point.*
*Correct. The original sentence was based on the implementation of the method and the assumptions used by Karion et al. (2015). We will clarify this in the revised manuscript and then draw the comparison to our data set which contains in situ wind measurements, allowing the calculation you suggest (which is now included in the new Table 3).*

l. 119: The cylinder pattern should be "around" a source and not only "near".
*You're right. We will correct it in the revised manuscript.*

l. 126: Why are the additional point sources considered sources of uncertainty?
*If there are not included inside the oval, any uncounted point sources could be considered additional "unknown" sources of GHGs that should have been included in our urban-scale study. "Uncertainty" was probably not the best word to use, and we will fix or omit this sentence in the revised manuscript.*

l. 131: I think we all got the concept of three-dimensional space (delete the parentheses).
*Yes, we will delete the parentheses.*
l. 134: The PBLH is not hard to measure. It is relatively easily determined from a vertical profile of temperature and humidity as you have done in this study. You even stated that the different approaches you used led to similar results. So what is difficult with respect to the PBLH? It is certainly difficult to model correctly, as you stated that substantial differences exist between models and reanalysis data. Also consider the large diurnal variability of PBLH.
*We will remove "that is hard to measure".*
l. 135: Please don't use "observed" in connection with models. This might confuse.
*Yes, we will correct it.*
l. 139: Do you really think the execution of flights is a goal of the study? Or is it merely necessary for the other goals?
*One of the goals of this study is to test the flux estimate when using the cylindrical flight pattern. To reduce the confusion, we will remove the word "execute" in the revised manuscript.*
l. 145: Which "value"?
*We will remove this in the revised manuscript.*
l. 153: Describe the three flights here and mention figure 1. It is not mentioned at all in the text.
*Yes, we will do it.*
l. 154: How many whole-air standards do you use for calibration?
*We used two NOAA whole-air standards, a "high" and a "low". Plus we used these whole air standards to put accurate numbers onto secondary, synthetic standards, of which we have 5 or more of varying ranges. So they give us a good handle on the linearity of the instrument, across varying concentration ranges.*

l. 159: Take off time is not sufficient. Please give the total flight times in UTC. Using Pacific Standard Time and Pacific Daylight Time here needs more explanation on why you give take-off times in different ways.

*One flight was executed in the winter and one during Daylight Savings Time. We will re-write this section more clearly: Sampling occurred 21:10 – 22:00 UTC for the November flight (local standard time is UTC minus 8 h, 13:10 – 14:00 PST) and 20:55 – 21:45 UTC for the June flight.*

Sect. 2.2: Review order of sections: I would first discuss the interpolation method and then the extrapolation.
*We will consider this suggestion when constructing the revised manuscript. The significant restructuring we plan to do (shown above) may benefit from discussing the interpolation first, but we can't judge which is best until we invest the time in restructuring and re-ordering the figures.*

l. 163: Reformulate: "Because the lowest flight level was typically between 250 m and 380 m above the surface …"
*We will correct it as suggested.*
l. 165: Sentence needs restructuring: The "unmeasured values" lead to uncertainty whether or not a "well-mixed layer assumption" is made. Split sentence!
*Yes, we will work on that.*
l.168: Refer to Figure 2 here. There is no reference to it in the text.
*We will do that.*
l. 169: What do you mean by "elevated" plume? Is it lifted of the ground or are there large enhancements of the concentration?
*We mean the former: lifted off the ground.*

l. 170: How exactly do you derive the background level? What is the "lowest flight measurement"? Has it got anything to do with the "lowest flight level" which you use in the formula? Is the background only determined from the lowest flight level? Why are you talking about background at this point? It is a section on extrapolation to the ground. Do you also extrapolate the background values? What do X and t stand for?

*We apologize for the wording in the original manuscript at line 170. Our description of the "constant" method was unnecessarily confusing. The blue curves in Figure 2 show what we were trying to describe: at all altitudes below the yellow diamond (lowest flight level), the mixing ratio was presumed to be exactly equal to the value measured at the lowest flight level. Please also see the new figure below in supplementary materials. The bottom left panel shows more clearly how the values measured along the lowest flight level are assigned to all grid cells below the measurement altitude.*

*X is the given trace gas concentration, and t is the single parameter representing each point on the ellipse (eccentric anomaly)*

l. 172: Do you mean that the details of the method are described in Gordon et al. (2015)?
*Yes, that is what we mean.*

l. 173: How is the Gaussian distribution of the plume dispersion calculated?

*The Gaussian fit method is similar to the exponential fit method, except that the surface-sourced plume dispersion follows a Gaussian distribution function. For a given set of (x: height, f(x): GHG) pairs, we get the rate of change and the mixing ration at the surface ($C_{suf}(s)$) at each s parameter from the gaussian function,*

$$f(x) = C_{top}(s) + (C_{suf}(s) - C_{top}(s)) * \exp(-x^2 / 2s^2)$$

Sect. 2.3: Consider renaming the section to "Measurement interpolation".
*Thank you. That is a nice suggestion.*

ll.176-186: Should this be a separate section called: Projection of data to cylinder surface?
*Yes, we wrestled with that. It is a bit too small to stand alone, but it might make sense to do that in the revised structure.*

l. 182: What is Y?
*(X(t), Y(t)) is the each point on the ellipse represented by a single parameter (t, eccentric anomaly). So X refers to the longitude and Y refers to the latitude.*

l. 190: Refer to Fig. S4 as you show these differences there. Consider over plotting the measurements on the kriged and interpolated fields for better assessment of your result that kriging better captures individual plume features. What altitude range do the elliptical cylinder plots cover? Ground to PBLH? Please state in the figure caption.
*Thank you for the good suggestion. To better explain the different interpolation methods, we will incorporate Fig. S4 with Fig. 2 in the revised manuscript. We plot up to the highest measurement altitude for the elliptical cylinder plots. However, for computing the actual fluxes, we only integrate the fluxes from the surface (z=0) to the top of the PBLH. We will state this clearly in the figure caption in the revised main and supplementary material.*

ll. 214-227: Consider a separate section on uncertainties.
*This is a good idea. We will make this as a separate section. See our new outline for the revised manuscript (page 1-2).*

l. 216: Not only downwind interpolated values induce uncertainty. Upwind values as well.
*You're right. We will remove "downwind" from the sentence.*
l. 226: Add "observations" behind "direction".
*Yes, we will do this in the revised manuscript.*
l. 229: Remove "to the choice of background value and" because this is not the topic of this section. You do not investigate the wind characteristics but the treatment of wind measurements.
*We will do it in the revised manuscript.*
l. 230: Remove "In one"
*We will correct it.*

l. 231: What is "measured points"? How did you measure them?
*We meant the discrete measurement locations (lon, lat, height) at a given time obtained by aircraft. We will endeavor to find better wording in the revised version.*

l. 233: Stick to one tense (averaged, equaled).
*We will be careful in using tense in the revised manuscript.*

l. 233: "By assuming non-divergence, mass can be balanced." This is correct, but is this really what you need here?

*This is important. Because this is not divergent, the inflow and outflow are the same, and we can apply the mass-balance idea to our flux calculation.*

*l. 242: Do you assume PBLH to be constant during your flights? At what time during the flight did you measure the profile?*
*Yes, we assume that PBLH is constant throughout our flights. Sampling profiles occurred 21:10 – 22:00 UTC for the November flight (local standard time is UTC minus 8 h, 13:10 – 14:00 PST) and 20:55 – 21:45 UTC for the June flight.*

l. 243: Is the boundary layer "growing" during your flights? How do you know?
*No, we assumed that there is not sufficient time for change in the PBLH during our flights (less than 1.5 hours) We think the confusion comes from the word that we used. We will change it from "boundary layer growth" to "boundary layer height" in the revised manuscript.*

l. 248: How is C(t,z) determined? How can one point surround the top of the cylinder? How is the background defined here?
*We apologize for the grammar error. The sentence should have read "C(t,z) is the $CO_2$ concentration (g $m^{-3}$) at each point around the top of the cylinder (where z=h), and $C_{bg}(h)$…"*
*We used this formulation for each method (sensitivity test) of defining the background.*

l. 252: Is the entrainment calculated from the kriged data?
*Yes.*

l. 263: Flux is defined through a surface. Thus it cannot be "inside" the cylinder.
*You are right. Flux is defined through a surface. We will fix the grammar in this section in the revised manuscript.*

ll. 265-280: See my comment in the General Comments section on the use of a background value with the Gaussian divergence theorem. If you consider inflow and outflow, you do not need a background. In your formula, the result should be invariant to the value of background mixing ratio chosen if you consider positive contributions as outflow and negative contributions as inflow.
*Thank you for elaborating on this. This is also why we used the mass-balanced mean wind, so that influx mass and outflux mass are the same and the total flux estimate is not dependent on having an understanding of background mixing ratios. We will mention it in the revised manuscript.*

l. 291: Use present tense.
*We will change it in the revised manuscript.*
l. 295: Remove "concentration".
*We will change it in the revised manuscript.*

l. 299: How is the kriged estimate less arbitrary in an area far away from measured values? What assumptions is it based on? Is the state of the PBL (stable/unstable) taken into account?
*We believe kriging is less arbitrary because we have more constraints for formulating a kriged estimate. When we calculate fluxes based only on the measured data, without filling in the gaps between flight levels, the total flux estimate will obviously be much smaller than when we account for the entire surface of the cylinder using interpolated data. With an urban-scale cylinder (with a circumference on the order*

*of 100 km), it is impossible to map out the entire surface (~100 km$^2$) with dense measurements. Although kriging cannot be better than actual observations, it can be a good alternative to "mimic" actual data. We disagree with the reviewer's opinion that we solely rely on the kriging without an understanding of the data. We carefully performed the variogram analysis, and carefully chose the kriging parameters (sill, range, and nugget) based on the experimental and theoretical variogram obtained from the actual data we measured.*
*In contrast, other methods are solely based on general assumptions without the actual inspection of the existing spatial dataset.*

l. 300: You don not mention the Gaussian fit method depicted in Figure 2 at all.
*We mentioned it in section 2.2 (Line 172) in the original manuscript. The Gaussian fit method is similar to the exponential fit method, except that the surface-sourced plume dispersion follows a Gaussian distribution. See the explanation above. We will mention it more clearly in the revised manuscript.*

Sect. 3.1: What is the influence of the different choice of interpolation and extrapolation on the flux estimate? Here a table similar to Table one would be great.
*Thank you for a good comment. However, what we focus in this study is the impact of treatment of wind measurement and background on the flux estimate, not comparing different interpolation methods (although we mentioned these for completeness using Fig. 2). Furthermore, although we did show different GHG mixing ratio assumptions below the lowest flight level, we do not consider how to treat the wind below the lowest flight level. We may assume a constant wind speed and compute the flux for each of the extrapolation methods, but we are not sure how to interpret those values and we believe this will gives us additional challenges, leading to additional uncertainty in total flux estimates without understanding the physical meaning of the calculated values. Furthermore, we already mentioned that the difference of CO$_2$ estimate below lowest flight level could lead to the change of GHG concentration up to 20%.*

l. 304: Remove "gap of the".*We will change it in the revised manuscript.*
ll. 314-320: Please mark all the locations mentioned in the text on a map so the reader can confirm your statement.
*We will mark it in the revised manuscript.*
l. 325: Present tense.
*Yes, we will be careful about using consistent verb tense.*
l. 327: Please check "a farther".
*We will change it to "far".*
l. 330: Maybe use the last sentence of this paragraph as its first. Good introduction.
*We will restructure the sentences in the revised manuscript.*

l. 350: The PBLH you determine from the vertical profile might have an uncertainty of <1%, but is this value representative for the whole measurement area with this accuracy? What about changes over time and with the location? How does a less defined PBLH influence the uncertainty?
*We assumed that PBLH does not change during our 1.5 hour flight. The urban-scale area studied is approximately 20 km x 40 km with pretty uniform topography, thus we expect the PBLH to be the same throughout the sampled domain. However, we do acknowledge that a different estimate of PBLH can increase the uncertainty. Please see the response below.*

Fig. S6: Looking at your method of estimating PBLH there seems to be a possible error of more than 1 % as well. In Fig. S6d it becomes clear, that you use the 50 m averaged values for checking the gradients. Then you place it at the top of the layer with the highest gradient. Here it is visible, that this point is easily 40 m above the layer where a 20 m averaged profile would see the gradient. Thus your uncertainty is around 50 m, which would be almost 10 % for a PBLH of 600 m.

*That is a very good point. As you pointed out, the uncertainty of the PBLH can be up to 10% if we determine the PBLH based on the largest gradient of the vertical profile of the potential temperature. We will consider the uncertainty of the PBLH estimate and include it in the total uncertainty estimate in the revised manuscript. Based on our 3 measurements, the uncertainty due to PBLH estimate for urban scale is about ~10% , but the uncertainty due to PBLH estimate for the local-scale is about 1-5 % so that the change of PBLH does not affect the total flux estimate. As seen in Fig. S6, the vertical range of the largest gradient of potential temperature is very small, compared to the urban-scale. This leads us to another important message: the uncertainty gets larger when we deal with urban-scale flux estimate. We will include the uncertainty due to the estimate of PBLH in the total flux uncertainty estimate in the revised manuscript.*

l. 355 ff: See my comment on the treatment of "mass-balanced wind" in the General Comments section.
*We made new tables for the comparison as you suggested. Please see the tables (page 2-3).*

Sect. 3.3: Please already refer to your Table 1 when naming the results.
*Yes, we will do so.*

l.267: Where is an "actual" location of the rice field? Pleas show locations on a map rather than just giving coordinates. This is very hard to visualize for a reader.
*The labels in Figure S8 are awfully hard to read, and for that we apologize. We will improve them in the revised version*

l. 370: "the local emissions are attributed to these high flux estimates". Did you mean: "The high flux estimates are attributed to the local emissions"?
*Yes. Thank you for pointing this out. We will fix it in the revised manuscript.*

l. 374: Formulation: "mean wind vector at the dominant wind direction (positive and one direction) and speed". How is this calculated?
*Many previous studies use the mean wind averaged over the PBLH. Karion et al. (2015) estimate the total $CH_4$ emission in the flight region (curtain flight) using a mass balance approach. According to their study, when the mean horizontal wind speed and direction are steady during the transit of an air mass across an area, the resulting calculated horizontal flux is equal to the surface emission between the background location and the downwind measurement. This calculation required the assumption of steady horizontal wind direction, a well-developed convective PBL, and measurements sufficiently downwind of the emission source such that the emissions are vertically distributed throughout the PBL.*

l. 381: There is no Table 2.
*We will fix this and refer to the right one in the revised manuscript.*
l. 387: Raw wind is displayed in the bottom two lines.
*We will fix it in the revised manuscript.*
ll. 390-391: This sentence is incomplete and not logical.

*We will change it in the revised manuscript.*

l. 394: Table 1 shows a range of 3.68 - 26.58 Mt $CO_2$ yr-1 for the whole city.
*Thank you for catching that. It must have been a hold-over from an earlier draft.*

l. 396 ff: Here you investigate the difference between using the complete ellipse and only the downwind part. This should be a separate section, and the results should be presented in another table.
*Yes, we will present it in a new table (Table 3) in the revised manuscript.*

l. 399: Change "From this study,…" to "According to these calculations…"
*Thank you for catching that. It must have been a hold-over from an earlier draft.*

l. 401: Table 1 gives a range of 13-92 Mt $CO_2$ yr-1 for Nov. 18, 2015.
*We will correct it in the revised manuscript.*

l. 402: Please indicate "Region-3" on a map.

[Figure]

*This is from the study led by Jeung et al. (2016), and region 3 refers to the Sacramento valley. Each number represents the region classification based on California Air Basins (https://www.arb.ca.gov/ei/maps/statemap/abmap.htm).*
*Thus, this covers much larger area than we actually measured for the flux calculation for this study. We will explain this better in the revised version.*

l. 405: Is vi) the same as i)?
*We will correct that.*
l. 405: Which of these does "This" refer to?
*We will work on the wording in the revised manuscript.*
ll. 415-422: "Note … Table 2)." All this is repetition to before and not about the topic of this section which is "vertical mass transfer".
*We will remove this part in the revised manuscript.*

l. 428: Remove "First,"
*We will remove this in the revised manuscript.*

l. 431: Specify: "different flux calculation methods"
*We will remove it.*

l. 453: There is a contradiction here "the final flux estimates become similar", because the beginning of the sentence states that the background value is a major source of uncertainty.
*As you pointed out, background concentration is not important for the cylindrical flight, and we actually showed that the total flux is insensitive to the choice of background concentration when we used the mass-balanced mean wind. We stated that background value is a major source of uncertainty when we do not use mass-balanced wind for cylindrical flights. We will rewrite this more clearly in the revised manuscript.*

l. 459: Insert "that" after "suggesting".
*We will insert it in the revised manuscript.*

ll. 460-468: This section is a general overview of the flight results and should be placed earlier in the Conclusions.
*We will move this section to the earlier in the conclusions.*

l. 463: An overview of wind conditions should also be placed in the Results section.
*This is a good point and we will find an appropriate place in the revised manuscript.*

l. 464: This result (isolated high concentrations of $CO_2$) has not been shown in the Results section either.
*We showed a high concentration of $CO_2$. This is shown in Fig. 4(c) and Fig. 5(a, d) in the original manuscript. We will more clearly discuss these plots in the revised manuscript.*

l.470: Why did you expect sources to be concentrated on the downwind side?
*We didn't mean that we expect sources to be on the downwind side. What we tried to state is that horizontal flux is transported to the downwind side. We think this confusion comes from the unclear wording. We are sorry for the confusion and we will be more clear in the revised manuscript.*

l. 471: "Furthermore" does not fit here.
*We will work on this in the revised manuscript.*

l. 471: Wind variability definitely influences the flux estimates, not only during different times of the year. So this seems logical. It would be much more interesting how large the uncertainty due to this is assumed to be.
*We agree with that. We were interested in the influence of the wind treatment on the final flux estimate. However, we only used the measured wind, not any other source of wind data. It would be interesting to understand the magnitude of the uncertainty of the total flux estimate depending on the source and treatment of the wind data (e.g. measured wind vs. modeled wind with different temporal and spatial resolutions), but this is beyond the scope of this study and will be a topic of the future study.*

l. 475: The size of the ellipse is another factor that appears here for the first time in the manuscript. There is no data given on how large your ellipses were and what the influence is in the Results section.

*We analyzed two flight data at different size – urban scale (~ 20 x ~40 km) and local scale (< 3km). We mentioned the scale in the introduction (Line 144 in the original manuscript), and it sounds like we should reiterate it in the new Section 2.1.*

l. 480 and 481: Remove two of the three "further".
*We will remove this in the revised manuscript.*
l. 482: Do you really want to assess: "seasonality of sensitivity of emission estimates"? Just start with the seasonality of emissions first.
*We will correct it in the revised manuscript.*
l. 484: Where do you show the sensitivity of emission estimate uncertainty to temperature and potential temperature?
*We will revise the sentences.*
ll. 490-491: This sentence needs some revision and focus.
*We will work on them in the revised manuscript.*

**Figures**
Fig. 1: There is no shading visible in Fig. 1c.
*By shading we meant "color fill" in Fig. 1c. The blue color represents inflow (airflow passing through the cylinder, negative sign) and red color represents outflow (airflow passing out from the cylinder, positive sign), respectively.*

Fig. 2: What is the "altitude of the lowest flight data"? Please indicate the location of these measurements on a map, giving coordinates is not very helpful.
*The altitude of the lowest flight can be shown in the time series plots (shown in the response to reviewer #1). We will include this in the revised manuscript. We will also indicate the location of the measurement in the map.*

Fig. 3d: Is the ellipse shown from the ground to the highest flight level or which altitude range?
*Yes, the ellipse is shown from the ground to the highest flight measurement level (~ 1000 m). This is the same as shown in the time series plots (shown in response to reviewer #1).*

Fig. 4: Please provide headings with the date of the flight for the left and the right column.
*Fig 4 will be significantly revised.*

Fig. 5: Why is the mean wind kriged? This has not been mentioned in the text.
*We first kriged the measured wind and then computed the mean wind (averaged the kriged wind) at each level. We tested levels of 100m, 200m, 500m, 1000m thickness, as well as the whole cylinder.*

Fig. 7: Why is there this large space between the two sets of bars? What is "area-mean"?
*This just means "mass-balanced mean wind (whole vertical layer)". We will be careful and consistent with our terminology.*

Table 1: Tables normally have their description above not below them.
*We will modify them in the revised manuscript.*

**Supplementary Material:**

There appear to be bits and pieces of text strewn throughout the Supplement. Please give them a heading and a number so it becomes clear where they belong, and you then may also refer to them from the main manuscript.
Fig. S1: Figure b color bar label is missing.
*We will correct it.*

l. 7: I am not sure if you can say that emissions are "accumulated" downwind. They are transported downwind, but accumulation would mean that there is very slow wind only.
*We agree with you. We will change the wording in the revised manuscript.*

ll. 11-12: This is not true. With a curtain flight it is also possible to detect emissions from more than one point source within the city, throughout the city and downwind. It gets problematic if there are sources further upwind of the city that gets mixed with the city plume and cannot be separated from it.
*We will correct them in the revised manuscript.*

ll. 15: You mention three types of flight patterns in the main manuscript but only show two of them here.
*We will carefully examine this in the revised manuscript.*

Fig S2: Reformulate "throughout the altitude". Color bar labels are missing.
*We will work on them in the revised manuscript.*

l. 25: "accumulated" s.a.
*We will work on them in the revised manuscript.*

l. 28: Why is air at lower wind speeds less dispersive?
*We will remove November 17, 2015 case, so this sentence will be significantly revised.*

l. 28 ff: Reformulate sentence "Both flights …"
*We will remove November 17, 2015 case, so this sentence will be significantly revised.*

l. 30: Who uses continental scale wind for flux estimates?
*We will remove the wind rose plots.*

Fig. S3: This figure is not mentioned in the main manuscript. Please add flight dates to the left of the plots.
*We mentioned this figure in Line 144 in the original manuscript, but we did not fully explain these plots in the main manuscript. Yes, we will add flight dates to them in the revised version.*

l. 36: Consider "falling". For methane the dashed line is blue. Remove "observation".
*We will work on them in the revised manuscript.*

Fig. S4: Why are there "boxes" or vertical cuts visible in (d)? Does this have to do with gridding? What is the grid size? Could you plot the measurements on top of the interpolated fields? This way it is easier to assess your statement "kriging reflects the individual plume characteristics better". Could you show the extrapolated fields to the ground as well? Which step is performed first: interpolation or extrapolation? Is this described in the text?
Also: Use the same color bar range for all plots.

*Yes, the boxes (vertical columns) are related to the bin size for the interpolation and fit. Above the lowest flight level, we can use interpolation, but below the flight level, we need to do extrapolation. The white boxes represents no result due to the lack of the number of data used.*

[Figure]

*The figures above show the $CO_2$ field extrapolated to the ground. We do both interpolation and extrapolation in one process. We applied a formula for Gaussian Fit and exponential fit (Gordon et al., 2015) based on the lowest flight level data). For the interpolation, we used the exponential weighting function for the data above lowest flight level (for the interpolation), and then a constant value for the locations below the lowest flight level (for the extrapolation). Yes, we will use the same color bar range for all panels. The black dashed line represents an approximate lowest flight line.*

l. 57: Don't (b) and (d) also show only the subset of the ellipse? Could you change the direction of these plots? Then this arrow would not be necessary.
*The horizontal range is much larger than the vertical range. So, it is very hard to see the actual difference if you try to compare the whole ellipse. That is why we just try to show only a subset of the ellipse for comparison. But as you see in the plots above, there is still a noticeable difference between kriging and interpolation with an exponential weighting function.  Yes, I did change the direction of this plot. The modified plot is below.*

[Figure]

l. 82: Remove the sentence: "The $CH_4$ enhancement was localized near the landfill." This is obvious.
*We will remove it in the revised manuscript.*
l. 83: Also remove "…, and we … case." This is also obvious.
*We will remove it in the revised manuscript.*
Fig. S7: This figure is not mentioned in the main manuscript.
*We will discuss it in the main manuscript in the revised version.*

**Reference**

*Conley, S., I. Faloona, S. Mehrotra, M. Suard, D. H. Lenschow, C. Sweeney, S. Herndon, S. Schwietzke, G Petron, J. Pifer, E. A. Kort, and R. Schnell: Application of Gauss's theorem to quantify localized surface emissions from airborne measurements of wind and trace gases, Atmos. Meas. Tech., 10, 3345-3358, https://doi.org/10.5194/amt-10-3345-2017, 2017.*

*Gordon, M., Li, S.-M., Staebler, R., Darlington, A., Hayden, K., O'Brian, J., and Wolde, M.: Determining air pollutant emission rates based on mass balance using airborne measurement data over the Alberta oil sands operations, Atmos. Meas. Tech., 8, 3745-3765, doi:10.5194/amt-8-3745-2015, 2015.*

*Jeong, S., et al.: Estimating methane emissions in California's urban and rural regions using multi-tower observations, J. Geophys. Res. Atmos., 121, 13,031–13,049, doi:10.1002/2016JD025404, 2016.*

*Karion, A. et al.: Methane emissions estimate from airborne measurements over a western United States natural gas field, Geophys. Res. Lett., Vol. 40, 1-5, doi:10.1002/grl.50811, 2013.*

*Karion, A. et al.: Aircraft-Based Estimate of Total Methane Emissions from the Barnett Shale Region, Environ. Sci. Technol. 2015, 49, 8124-8131, DOI: 10.1021/acs.est.5b00217, 2015.*

*Kountouris, P., Gerbig, C., Rodenbeck, C., Karstens, U., Koch, T. F., and Heimann, M., 2018: Atmospheric $CO_2$ inversions on the mesoscale using data-driven prior uncertainties: quantification of the European terrestrial $CO_2$ fluxes.*

*Nathan, B., Golston, L. M., O'Brien, A. S., Ross, K., Harrison W. A., Tao, L., Lary, D. J., Johnson, D. R. Covington, A. N., Clark, N. N., and Zondlo, M. A.: Near-Field Characterization of Methane Emission Variability from a Compressor Station Using a Model Aircraft, Environ. Sci. Technol, 2015, 7896-7903, DOI: 10.1021/acs.est.5b00705, 2015.*

*Peylin, P., Rayner, P. J., Bousquet, P., Carouge, C., Hourdin, F., Heinrich, P., Ciais, P., and AEROCARB contributors, 2005; Daily $CO_2$ flux estimates over Europe from continuous atmospheric measurements: 1, inverse methodology, Atmos. Chem. Phys., 5,2173-3186, 2005.*

*Turnbull, J. C., Karion, A., Fischer, M. L., Faloona, I., Guilderson, T., Lehman, S. J., Miller, B.R., Miller, J. B., Montzka, S., Sherwood, T., Saripalli, S., Sweeney, C., and Tan, P.P.:: Assessment of fossil fuel carbon dioxide and other anthropogenic trace gas emissions from airborne measurements over Sacramento, California in spring 2009. Atmos. Chem. Phys., 11, 705–721, 2011, doi:10.5194/acp-11-705-2011, 2011.*

---

## Editor Comment (EC1) · Kiemle (Editor) · 21 Dec 2018

Dear Ju-Mee Ryoo, your answers to the ample reviewers' comments are substantial. I am convinced that they will considerably improve your paper and therefore I encourage you to submit a revised manuscript. Since both referees have requested major revisions I will ask them to carefully assess the new version. Best regards, Christoph Kiemle

---

## Referee Report (RR1)

**Review of Revised Submission:**

Quantification of CO2 and CH4 emissions over Sacramento, California based on divergence theorem using aircraft measurements: Ju-Mee Ryoo et al.

**General Comments:**

After the revision this manuscript has improved tremendously. The structure is much clearer and results are presented more uniformly. Figures and tables are well presented and include all the necessary information for the conclusions. Furthermore, a lot of issues due to language use have been resolved. Still, I have two minor revision points I would like to be addressed clearer before publication:

First, I believe that your treatment of interpolation/extrapolation methods is not complete. You show different methods and compare them, but never show how they influence the final emission estimate. A small section on "Sensitivity of the calculated flux to interpolation/extrapolation method" would be great at the beginning of chapter 3.2.

Second, the chapter on uncertainties needs restructuring and clarification. Several factors of uncertainty are named at several locations in the text, e.g. wind (I. 415, I. 423, II. 441-446) and kriging (II.411-414 and II. 425-429). Please consolidate these sections for better readability. Your method (II. 430-432) is mentioned after some factors of some uncertainty, but afterwards (I. 432 ff) you start listing additional factors (PBLH, vertical fluxes) again. Please list all factors first, or present your method first, but don't mix. How do you get to the overall estimate of 10% uncertainty? What is the variance and standard deviation of the kriged mixing ratios? Grid resolution (4%), variogram model (5%), wind measurement (? %), mixing ratios (? %), PBLH (10 %), vertical fluxes (1%), does not sum up to 10%. Please explain.

Additionally, how do your sensitivity studies influence your uncertainty estimate? You do all these sensitivity studies, receiving differences in the emission estimates of up to 80 %. Is that no uncertainty?

**Specific Comments and Technical Corrections:**

I. 49: "The uncertainty is also impacted by meteorological conditions and distance from the emission sources". These are two factors that you don't discuss in Chapter 3.3.

I. 50: "The largest CH4 mixing ratio was found over a local landfill." Is this a key finding of your study? Should I be in the abstract at this position?

I. 58: "modeling" might confuse here. You did not do any real modeling.

I. 61: "identifying emission sources". Is that what you focus on in this paper?

I. 66: Is air quality the main concern of this paper? GHG are generally not considered as air pollutants.

I.143: Here some sentences are different from the marked-up version of your manuscript.

I. 149ff: Here a whole section is not in your marked-up version. Which will be the final version?

I. 315: You should investigate this huge difference in you emission estimates due to wind treatment a little more. Could the cause be a certain meteorological condition? Maybe an accumulation of methane and CO2 in the outflow region due to low wind speeds. If you then average the wind over the entire loop, it might be extremely overestimated in the outflow region and the flux is too high. What is the difference between raw wind and mass-balanced wind in relation to GHG mixing ratio along the lowest and only flight track within the boundary layer?

I. 331: As visible from Fig. S5 only your lowest ellipse lies within the PBL. Clearly the wind is quite different within the PBL from above. Does whole-column-averaged wind (as in Fig. 6c) make sense in this case?

I. 339: "Flux estimates using raw wind are more sensitive to the choice of background..." Do you have any idea why it is this way?

I. 388: Table 3 states 13.4 Mt CO2 for Turnbull.

I. 395ff: Which areas do the Turnbull et al (2011), Vulcan, and CEPAM inventory values given here cover?

I. 396: Please give a reference for the 1.1% annual increase in CO2 fluxes.

I. 401: Are there any CH4 emission inventories you could compare your emission estimates with? How about EDGAR or the U.S. Environmental Protection Agency's inventory?

II. 467-479: Here you emphasize how the ellipses flight pattern reduces the sensitivity towards the choice of background. But, using a curtain flight the choice of background only induces 50 % difference, while for an ellipses flight the difference between wind-treatment techniques is 80%. How does this relate? How sensitive are curtain flights to wind treatment? Please add in Table 3. Why would you still recommend ellipses flights?

I. 493: "Second, the seasonality ..." This sentence does not make sense.

**Figures:**

Figure 3: What does the gray dashed line in (c)-(f) depict?

Figure 6: It is hard to believe that between case (a) and (b) the difference is 86 %, but between case (b) and (c) only 3.9 %. Case (c) looks as if the flux estimate should be much larger than (b). Do you have an explanation?

Figure 7: Again it is hard to understand from the figures how (d) and (e) differ by 25 % but (e) and (f) only 2.8 %. The flux in (d) and (e) look almost the same, while (f) looks much different.

**Tables:**

Table 1 and 2: Please indicate negative differences from the base case with a minus in front of the given percentage.

Table 3: Please add a row of results for curtain flights using raw wind.

---

## Author Response (AR2)

**Ms. Ref. No.: AMT-2018-254**

**Title: Quantification of CO2 and CH4 emissions over Sacramento, California based on divergence theorem using aircraft measurements**

*Dear Dr. Christoph Kiemle,*

*Thank you very much for your efforts on behalf of our paper titled "**Quantification of CO$_2$ and CH$_4$ emissions over Sacramento, California based on divergence theorem using aircraft measurements**" submitted by Ju-Mee Ryoo, Laura T. Iraci, Tomoaki Tanaka, Josette E. Marrero, Emma L. Yates, Inez Fung, Anna M. Michalak, Jovan Tadic, Warren Gore, T. Paul Bui, Jonathan M. Dean-Day, Cecilia S. Chang.*

*We found the reviewers' comments helpful, and we have addressed the reviewers' comments and suggestions and incorporated them in the revised manuscript. Enclosed is a point-by-point response to the review comments.*

*Thank you very much again for your support, and we look forward to hearing a positive decision from you soon.*

*Sincerely,*

*Ju-Mee Ryoo and Coauthors*

**Response letter to Reviewer #1**

*Responses from the Authors are given in blue italicized text throughout: Thank you for your encouragement and helpful comments. We addressed them below.*

I would like to thank the authors for making substantial revisions to their manuscript and improving it significantly. I will accept the publication of this article after a couple of minor revisions.

Paragraph starting on line 430: Please discuss uncertainty due to background in some detail. From my experience it could contribute significantly to the overall error. Please mention this part somewhere in this paragraph or section.

*We discussed uncentainty due to the background in section 3.3.*

Figure 4: please add wind direction as you did in Figure 5.

*Thank you for the suggestion. We added wind direction in Fig. 4 too as you suggested.*

[Figure]

*Figure 4: (a) Map of AJAX flight tracks colored by CH₄ mixing ratio for November 18, 2013, plotted in Google™ Earth. (b) The data (red) fitted to an oval (green). The observed CH₄ mixing ratios (c) are kriged to generate the cylindrical surface (d). The axes of the oval are approximately 25 and 40 km. The yellow arrow represents the dominant wind direction.*

**Response letter to Reviewer #2**

*Responses from the Authors are given in blue italicized text throughout: Thank you for your encouragement and helpful comments. We addressed them to the best of our understanding and appreciated your guidance on areas where you found our discussion and plots unclear.*

**Review of Revised Submission:**

Quantification of CO2 and CH4 emissions over Sacramento, California based on divergence theorem using aircraft measurements: Ju-Mee Ryoo et al.

**General Comments:**

After the revision this manuscript has improved tremendously. The structure is much clearer and results are presented more uniformly. Figures and tables are well presented and include all the necessary information for the conclusions. Furthermore, a lot of issues due to language use have been resolved. Still, I have two minor revision points I would like to be addressed clearer before publication:

First, I believe that your treatment of interpolation/extrapolation methods is not complete. You show different methods and compare them, but never show how they influence the final emission estimate. A small section on "Sensitivity of the calculated flux to interpolation/extrapolation method" would be great at the beginning of chapter 3.2.

*As suggested, we added the section. Please see it in Section 3.2.1 in the revised manuscript.*

Second, the chapter on uncertainties needs restructuring and clarification. Several factors of uncertainty are named at several locations in the text, e.g., wind (l. 415, l. 423, ll. 441-446) and kriging (ll.411-414 and ll. 425-429). Please consolidate these sections for better readability. Your method (ll. 430-432) is mentioned after some factors of some uncertainty, but afterwards (l. 432 ff) you start listing additional factors (PBLH, vertical fluxes) again. Please list all factors first, or present your method first, but don't mix. How do you get to the overall estimate of 10% uncertainty? What is the variance and standard deviation of the kriged mixing ratios? Grid resolution (4%), variogram model (5%), wind measurement (? %), mixing ratios (? %), PBLH (10 %), vertical fluxes (1%), does not sum up to 10%. Please explain.

*Thank you for the suggestion. We rewrote the paragraph for the uncertainty section in the revised manuscript and believe this better identifies the terms that are summed in quadrature to determine the overall uncertainty. The estimated kriging variance (its square root is the standard deviation) is a natural byproduct of the kriging. It gives the estimation error, along with the actual estimate of the variable itself.*

Additionally, how do your sensitivity studies influence your uncertainty estimate? You do all these sensitivity studies, receiving differences in the emission estimates of up to 80 %. Is that no uncertainty?

*This is a good point. An uncertainty analysis is not the same as a sensitivity analysis. As with a more traditional "error analysis" for the measurement of a physical property, we view **uncertainty** as a*

*quantitative understanding of the impact on the final result of the numerical assignments we made for quantities such as PBLH and the kriged values for $CO_2$, $CH_4$, and wind. Beyond this "calculation uncertainty," we are fully aware that fundamental choices in our method (such as how to select the background or whether to use raw or averaged winds) will influence the resulting fluxes. Because these choices do not lend themselves to standard deviations or even "replicate measurements" (as one could consider the analysis we used for uncertainty in PBLH), we consider studies of the influence of these fundamental choices to be tests of the **sensitivity** of the outcome to the choice made in each category.*

**Specific Comments and Technical Corrections:**

l. 49: "The uncertainty is also impacted by meteorological conditions and distance from the emission sources." These are two factors that you don't discuss in Chapter 3.3.

*Thank you for pointing them out. I have incorporated them into the revised manuscript.*

l. 50: "The largest CH4 mixing ratio was found over a local landfill." Is this a key finding of your study? Should I be in the abstract at this position?

*This can be one of important findings of our study, but we don't think this finding necessarily needs to be in the abstract. So we removed it from the abstract. We think the impact of the difference in shapes of flight, wind treatment, vertical mass transfer, and background to the flux estimate is most important key messages of this study.*

l. 58: "modeling" might confuse here. You did not do any real modeling.

*This is a good point. We removed this in the revised manuscript.*

l. 61: "identifying emission sources". Is that what you focus on in this paper?

*This is not a focus of this paper, but we think this study showed that a closed-shape flight pattern could be a useful strategy for identifying local emission sources (e.g., landfill) and estimating local-scale greenhouse gas emission fluxes too. In that sense, we think we can write this in the abstract.*

l. 66: Is air quality the main concern of this paper? GHG are generally not considered as air pollutants.

*Good point. Air quality is not the main concern of this paper, but GHG can indirectly affect air quality too because GHG such as $CO_2$ and $CH_4$ can result in climate change. Reversely, atmospheric warming associated with climate change has also the potential to increase ground-level ozone and other air pollutants, such as particulate matter. That is why we mentioned it in the introduction.*

l.143: Here some sentences are different from the marked-up version of your manuscript.

*We accidently didn't change that part. We updated in the revised manuscript.*

l. 149ff: Here a whole section is not in your marked-up version. Which will be the final version?

*We update the manuscript many times, so this has not been reflected in the marked-up version. Sorry about that. The order of sentences has been changed while we edit the manuscript (e.g., Sampling*

*occurred….. in the marked-up version went into the last of the first paragraph of section 2.1). The revised manuscript is the correct one.*

l. 315: You should investigate this huge difference in your emission estimates due to wind treatment a little more. Could the cause be a certain meteorological condition? Maybe an accumulation of methane and CO2 in the outflow region due to low wind speeds. If you then average the wind over the entire loop, it might be extremely overestimated in the outflow region and the flux is too high. What is the difference between raw wind and mass-balanced wind in relation to GHG mixing ratio along the lowest and only flight track within the boundary layer?

*The differences in our flux estimates can be explained with different factors as you suggested. First, the huge difference in our emission estimates can be due to certain meteorological condition. The raw wind, in that case, may not represent the wind that carries pollutant downwind region. We can tell that wind speeds get lower when it is near the surface. In that case, the flux in the outflow region can be small because of low wind speed. Note that when we average the entire loop, we already only consider the data below the planetary boundary layer height (PBLH).  And when we average the wind over the entire loop below the PBLH, the wind coming into the cylinder (inflow) is negative and the wind out of the cylinder (outflow) is positive.*

*Second, the difference in our emission estimates can be due to the size of scales and distance to the source regions (e.g. urban or local scale). As seen in Table 2, the difference in emission estimate using different methods becomes smaller at the local scale.*

*The difference in our emission estimates can vary among flights too with those reasons above. However, the key point that we want to deliver here is that using mass balance wind along with closed-shaped flight can reduce the uncertainty of choice of wind and background, further to the emission estimate.*

331: As visible from Fig. S5 only your lowest ellipse lies within the PBL. Clearly the wind is quite different within the PBL from above. Does whole-column-averaged wind (as in Fig. 6c) make sense in this case?

*We thought we mentioned it clearly in the manuscript, but it seems that it wasn't clearly stated. The whole-column-averaged wind here means vertically averaged wind up to the estimated "PBLH". We wanted to show all the measured data (from the lowest flight level up to roughly 1000 meter for urban scale and 2000 meter for local scale vertically), but we used data up to the estimated PBLH for actual flux calculation. We have revised Figures 6 and 7 to reduce the confusion.*

l. 339: "Flux estimates using raw wind are more sensitive to the choice of background…" Do you have any idea why it is this way?

*One of the advantages of applying the mass balance approach with an oval (enclosed shape) flight path is that an assignment of the background concentration is not required. Because we calculated the mass of the GHG that entered and exited the cylinder, we are subtracting all GHG upwind from all GHG downwind. So, the background should cancel out. This is also why we used the mass-balanced mean wind, so that influx mass and outflux mass are the same and the total flux estimate is not dependent on having an understanding of background mixing ratios.*

*However, we do need to know the background value for estimating flux when adopting the curtain pattern of flight. Therefore, we stated that flux estimate using raw wind is sensitive to the choice of the background value (when we do not use mass-balanced wind for cylindrical flights). Sometimes, actually in most cases, the background is not homogeneous, so there can be large uncertainty associated with calculating the background when the non-mass balanced, raw wind is used. There are also challenges in isolating the plume. In a mass-balanced wind approach, such problems do not exist.*

l. 388: Table 3 states 13.4 Mt CO2 for Turnbull.

*Thank you for pointing this out. That was a typo, and we corrected it in the revised manuscript.*

l. 395ff: Which areas do the Turnbull et al. (2011), Vulcan, and CEPAM inventory values given here cover?

[Figure]

*The figure above shows the area where Turnbull et al. (2011) measured. The two bottom-up inventory estimates Vulcan and CARB CEPAM database of the annual total emissions are obtained from Sacramento County.*

l. 396: Please give a reference for the 1.1% annual increase in CO2 fluxes.

*Turnbull et al. (2011) use this rate for their study, so we used that to make a consistent comparison with their results. We have included the citation in the revised manuscript.*

l. 401: Are there any CH4 emission inventories you could compare your emission estimates with? How about EDGAR or the U.S. Environmental Protection Agency's inventory?

*As you suggested, we looked at the 0.1° x 0.1° gridded 2012 national U.S. Environmental Protection Agency (EPA) CH4 emission data reported by Maasakkers et al. (2016). The November CH4 emission for our study region around Sacramento is about 89.7 (58.8 − 120.5) Gg yr$^{-1}$ by increasing from 2012 estimate to 2013 by applying a change of 0.6% per year for CH$_4$. We added it in Table 3.*

| | | CO2 (Mt yr$^{-1}$) | CH4 (Gg yr$^{-1}$) |
|---|---|---|---|
| **Whole cylinder–AJAX** | **(bg = min, 100m layer avg)** | 25.6 ± 2.6 | 87.1 ± 8.7 |
| | **(bg = avg, 100m layer avg)** | 25.6 ± 2.6 | 87.4 ± 8.7 |
| **Curtain –AJAX** | **(bg = min)** | 17.3 ± 1.7 | 64.4 ± 6.4 |
| | **(bg = avg)** | 8.9 ± 0.9 | 24.1 ± 2.4 |
| **Turnbull et al. (2011)** | | 13.4 (with uncertainty of ~ 100%) | |
| **Vulcan estimates for Sacramento** | | 11.5 | |
| **CEPAM estimate for Sacramento** | | 10.0 | |
| **Maasakkers et al. (2016) Gridded EPA data (November)** | | | 89.3 (58.5 – 120.0) |

ll. 467-479: Here you emphasize how the ellipses flight pattern reduces the sensitivity towards the choice of background. But, using a curtain flight the choice of background only induces 50 % difference, while for an ellipses flight the difference between wind-treatment techniques is 80%. How does this relate? How sensitive are curtain flights to wind treatment? Please add in Table 3. Why would you still recommend ellipses flights?

*For curtain flights, we need to be more careful about wind treatment and background choice. As seen in Table 3 (we added in the revised manuscript), the calculated flux gets more variable depending on the choice of background. The wind treatment will also add challenge to flux calculation. The closed-shape flight using mass-balanced wind will prevent concern.*

l. 493: "Second, the seasonality …" This sentence does not make sense.

*We changed it into "Second, the variability of emission estimates with season needs to be examined. "*

**Figures:**

Figure 3: What does the gray dashed line in (c)-(f) depict?

*They depict the rough lines of the lowest flight altitude. We added it in the revised manuscript.*

Figure 6: It is hard to believe that between the case (a) and (b) the difference is 86 %, but between case (b) and (c) only 3.9 %. Case (c) looks as if the flux estimate should be much larger than (b). Do you have an explanation?

*We apologize for the confusion by Figures 6 & 7. In the plots, we wanted to show all the data we measured during the flight, even above the boundary layer height (typically less than 1km above the surface). However, we used the data only up to the top of the estimated boundary layer height when we calculated the final fluxes shown in the Tables. In the old version, we had plotted the fluxes calculated using all the data, which is different than how we calculated the fluxes in the Tables. This was the source of your question about Figs. 6(a, b, c) (as well as Figs. 7(d, e, f)) looking different from their percentages. We updated Fig. 6 and Fug. 7 in the revised manuscript by showing the plots only up to the top of the*

*estimated PBLH to prevent this confusion. This has the added benefit of allowing a visual comparison of the figures to the tables.*

Figure 7: Again it is hard to understand from the figures how (d) and (e) differ by 25 % but (e) and (f) only 2.8 %. The flux in (d) and (e) look almost the same, while (f) looks much different.

*Please see above.*

[Figure]

*Figure: (a) The map of AJAX flight track with the observed CH$_4$ mixing ratio. (b) The observed CH$_4$ mixing ratio, (c) kriged CH$_4$ mixing ratio, (d) CH$_4$ flux using raw wind, (e) CH$_4$ flux using the mass-balanced wind with 100 m vertically averaged wind, and (f) CH$_4$ flux using mass-balanced wind (averaged through the entire PBLH) over the landfill location on July 29, 2015. The fluxes are computed based on equation (2). The background value was chosen as the minimum value at each vertical layer. The approximate diameter of the cylinder is 3 km, and the color scale is capped at 2.2 ppmv in panels (b) and (c). The black dashed line represents the top of the PBLH for this flight. Figs. (c-f) are only shown below the PBLH.*

**Tables:**

Table 1 and 2: Please indicate negative differences from the base case with a minus in front of the given percentage.

*Good points. We added negative differences from the base case with a minus in the revised manuscript.*

Table 3: Please add a row of results for curtain flights using raw wind.

*As suggested, we also computed the flux estimate over the downwind part (assuming this is for curtain flight) using raw wind. However, note that the fluxes are significantly small ($9.1\pm0.9$ Mt yr$^{-1}$ for CO$_2$ and $37.2\pm3.7$ Gg yr$^{-1}$ for CH$_4$) compared to those using the mass-balanced wind.*

*In the traditional mass balance analysis, the incoming mass should be the same as the outgoing mass passing the X-Y plane of the measurements. When we used the raw wind measurement, however, we found that the mass of air mass was not conserved, which means we may not fully apply the mass balance approach. The raw wind, in this case, may not represent the wind that carries pollutant downwind region. That is why we used the mass-balanced mean wind at each grid level, and the mean wind will distribute the greenhouse gas inside the cylinder. We assume that the well-mixed condition applies.*

*Therefore, we don't think using the raw wind at the measurement point is appropriate at this point especially for the urban case so that we don't think we ought to include that in the comparison table.*

**Reference**

[revised manuscript text omitted]